

# The design and characterization of a High-Performance Single-Particle Aerosol Mass Spectrometer (HP-SPAMS)

Xubing Du[1,3], Qinhui Xie[1,3], Qing Huang[1,3], Xuan Li[1,3], Junlin Yang[2], Zhihui Hou[2], Jingjing Wang[2], Xue Li[1,3], Zhen Zhou[1,3], Zhengxu Huang[1,3], Wei Gao[1,3], Lei Li[1,3*]

[1]Institute of Mass Spectrometer and Atmospheric Environment, Jinan University, Guangzhou, 510632, China
[2]Guangzhou Hexin Analytical Instrument Limited Company, Guangzhou, 510530, China
[3]Guangdong Provincial Engineering Research Center for on-line source apportionment system of air pollution, Guangzhou, 510632, China

*Correspondence to*: Lei Li (lileishdx@163.com)

**Abstract.** This study describes a high-performance single-particle mass spectrometry (HP-SPAMS) design in detail. The comprehensive improvements of the injection system, optical sizing system, mass spectrometry, and data acquisition system have improved particle detection efficiency and chemical analysis. The combination of an aerodynamic particle concentrator (APC) system and a wide range of aerodynamic lenses (ADLs) enables the concentration of particles in the 100-5000 nm range by a factor of 3-5 times. The ion delayed exaction technology of bipolar time-of-flight mass spectrometry improves the mass resolution by 2~3 times, allowing the differentiation of multiple homogeneous masses of different substances. Moreover, the 4-channel data acquisition technology greatly enhances the dynamic range of mass spectrometry. The as-improved HP-SPAMS enhances the overall capability of the instrument in terms of particle detection number and detection efficiency. Moreover, it improves accuracy and sensitivity for component identification of individual particles.

The experimental performance of HP-SPAMS shows that the sizing detection efficiency of polystyrene latex microspheres is almost 70 %-100 % in the range of 300-3000 nm. Compared to the previous SPAMS, HP-SPAMS has a larger inlet flow rate, detection efficiency, and higher laser frequency, which makes HP-SPAMS increase the effective number of particles detected in the air by 47.8 times and improve the temporal resolution of detection. For the analysis of individual particles, the HP-SPAMS's improved resolution helps distinguish between most organic fragment ions and metal ions and facilitates the analysis of complex aerosol particles. For the analysis of individual particles, the increased resolution of the HP-SPAMS contributes to the differentiation of most organic fragment ions and metal ions and facilitates the evaluation of complex aerosol particles. In the case of atmospheric lead-containing particles, for example, HP-SPAMS can completely differentiate the isotopes of lead elements and the number of lead-containing particles is 145 times higher than that detected by SPAMS. The outstanding detection efficiency and chemical analysis capability of HP-SPAMS will be of great importance for low concentration aerosol detection and complex aerosol component analysis.





## 1 Introduction


Atmospheric aerosols could impact atmospheric visibility, cloud formation, and rainfall, among others, and have been the focus of research on atmospheric physicochemical processes. Single-particle aerosol mass spectrometry (SPMS), an aerosol monitoring instrument (Murphy, 2007), can analyze information such as particle size and composition at the level of individual aerosols in real time. Moreover, it has the advantage of fast response time and high sensitivity, which has been

applied to investigate environmental climate and atmospheric chemistry. (Murphy et al., 2006; Zhang et al., 2015; Zhang et al., 2009) However, the limitations of current mass spectrometry (MS) detection capabilities render it not yet well suited for analyzing complex aerosol components in low concentration environments. One reason is that the low inlet flow rates and poor transport efficiencies could limit the number of aerosols that SPMS could analyze in low-concentration environments, such as high-altitude mountainous or the atmospheric stratosphere. (Jost et al., 2004; Schmidt et al., 2017) Although enough

particles could be obtained by long-time collection, the mixing state of atmospheric aerosol particles may change with the environment. (Sun et al., 2019) Thus, improving the collection capability of SPMS could be necessary for accurate analysis. Another reason is that the poor mass resolution and detection sensitivity of SPMS make it difficult to obtain an accurate analysis of aerosols. (Pratt et al., 2009; Zelenyuk et al., 2009)

The number of particles detected in real-time is closely related to the inlet flow rate, the particle transmitting efficiency, and

the particle detection efficiency (Li et al., 2011; Murphy, 2007; Noble and Prather, 2000; Pratt and Prather, 2012). The commonly used aerodynamic lens inlet systems have good particle focusing performance, but the transmitted flow rate limits the inlet flow rate.(Liu et al., 1995; Wang et al., 2005) Increasing the inlet flow rate could increase the number of particles detected per unit time in SPMS. Cahill et al. developed a high flow rate injection system by combining a virtual impactor with an aerodynamic lens (Cahill et al., 2014). Our team also developed a particle concentration device to increase the

injection flow rate by a factor of 5.  The transmission efficiency of the inlet system also affects the number of particles measured since some particles are lost in the critical orifice, buffer chamber, etc. (Chen et al., 2007; Hwang et al., 2015). At the same time, the small divergence of the aerodynamic lens could also lead to large transmission divergence of particles over long distances, which are difficult to be detected by the sizing/ionization optical system, especially for non-spherical particles. (Clemen et al., 2020; Murphy, 2007; Zelenyuk et al., 2009) The particles that could be transmitted to the

instrument are not entirely measurable, which could also be related to the particle sizing efficiency and ionization probability. The sizing efficiency could be improved by increasing the intensity of the collected scattered light signal(Zawadowicz et al., 2020) or by optimizing the optical path and selecting a more sensitive detector to improve the signal-to-noise ratio(Zelenyuk et al., 2009). Whether the particles could be ionized or not is not only connected to the particle focusing characteristics of the inlet system and the accuracy of the measurement of the sizing system but also related to the particle composition, the

maximum operating frequency, wavelength, and energy threshold of the ionization laser, etc. (Thomson et al., 1997; Zelenyuk et al., 2008)



The detection sensitivity and mass spectral resolution of SPMS directly affects its ability to analyze individual complex component aerosols. The sensitivity of single-particle detection is mainly related to the ion yield of the laser ionization of the particles. (Zawadowicz et al., 2015) Previous studies have shown that the ion yield and ablation depth of laser depend on the laser energy density (Cahill et al., 2015; Wenzel and Prather, 2004), with low energy resulting in a low threshold to be dissociated and high energy increasing the abundance of organic fragmentation. (Thomson et al., 1997) Therefore, the non-uniform Gaussian beam usually used in SPMS could make the particles fall at different locations and obtain different energies, resulting in different sensitivity responses of different particles. (Steele et al., 2005) SPMS typically uses a time-of-flight mass analyzer (TOF-MS) to investigate the different mass-to-charge (m/z) ions simultaneously. However, the laser ionization source could cause a large initial kinetic energy dispersion of ions (Vera et al., 2005), making it difficult for TOF-MS to compensate for the kinetic energy difference through the reflector of TOF-MS, resulting in a low mass resolution of the method. To improve the accuracy of qualitative analysis, the identification of ions in the spectra is improved by organic fragment ions, metal oxide ions, and isotopic ions (Tan et al., 2002). However, these methods have reduced applicability for ions with mass deviations less than 1 Da. Ions delay extraction could be an effective method to enhance the resolution of laser ionization mass spectrometry (Kinsel and Johnston, 1989). Our team has also developed an exponential form of ion delay extraction technique, which can effectively improve the performance of SPMS in terms of resolution and hit rate. (Chen et al., 2020; Li et al., 2018)

Although there are many studies to improve one aspect of the performance of SPMS, the instrument's overall performance is insufficient. In this study, a new high-performance single-particle aerosol mass spectrometer (HP-SPAMS) is developed to enhance instrument performance regarding the number of detected particles, transmission efficiency, resolution, and sensitivity. Firstly, the structure and design of the HP-SPAMS are described in detail. Then the detection capability of HP-SPAMS and SPAMS for the number of particles is compared and analyzed in terms of the efficiency of sizing and hit rate. Furthermore, the detection results of the as-configured system are shown. Finally, the improvement of resolution and sensitivity on the detection results of individual particles by HP-SPAMS and SPAMS are compared and examined.

## 2 Instruments and methods

HP-SPAMS was designed based on the SPAMS (Li et al., 2011), and its schematic structure is shown in Figure 1a. Briefly, the aerosol particles were introduced through an aerodynamic particle concentrator (APC) into an improved aerodynamic lens (ADLs) to focus into a particle beam. The particle size was measured by a scattering system with two higher-power continuous lasers. Chemical information of the particles was obtained by bipolar TOF-MS analysis of ions derived from pulsed laser ablation/ionization of particles. The photograph of the HP-SPAMS setup is shown in Figure 1b. The instrument's main body was placed in the upper part of the frame, and the lower part was composed of the power supply, control unit, and pump. The size of the whole machine was 960 mm × 740 mm × 1000 mm (excluding the height of the inlet module) and weighed about 220 Kg.



### 2.1 Aerosol inlet and sizing system

To increase the inlet flow, HP-SPAMS increased the critical orifice diameter from 0.1 mm to 0.18-0.22 mm, increasing the inlet flux to 0.3-0.5 L/min. However, the surplus flow could affect the focusing effect of particles in the aerodynamic lens and couldn't maintain a suitable vacuum. A scroll pump (IDP-3 Agilent) was supplied at the front of the aerodynamic lens to extract the surplus flow. A separation cone was added below the critical orifice, so the particles were not pumped away with the surplus flow (Figure 1). Due to the difference in inertia between the gas and the particles, most particles have entered the

separation cone. This structure is like the virtual impactor, which could achieve the concentration of particles, so it is called an aerodynamic particle concentrator (APC). In addition, to analyze large particle sizes, the ADLs were designed (Wang et al., 2005; Wang and McMurry, 2006) and optimized using Fluent software, and the original aerodynamic lens apertures were optimized to 5.0 mm, 4.8 mm, 4.4 mm, 4.1 mm, and 3.9 mm. The optimized and improved inlet system could theoretically transport particles in the range of 100-5000 nm.

Two continuous wave Nd:YAG (532 nm) laser beams (500 mW, LaserWave, LWGL532 nm-500 mW) orthogonally spaced at 6 cm from each other irradiated the particles to produce scattered light, focused by an ellipsoidal mirror and then detected by a photomultiper tube (PMT H10721-110, Hamamatsu). Compared with SPAMS, increasing the laser power enhanced the intensity of the scattered light from the particles. Furthermore, by optimizing the optical path system and low filtering (1.9 MHz) of the signal of PMT, the background noise level was effectively reduced, and the detection capability for small-

size particles was improved. The whole sizing system was calibrated with standard polystyrene latex (PSLs) material, and the formula $d_a = a + bt + ct^2 + dt^3$ was used for particle sizing to meet the accuracy of a wide particle size range. In addition, the HP-SPAMS sizing system could work in the conventional sizing and counting mode but also add a new compensation model. This mode increased the number of detections of small particles by periodically selecting particles of different particle size segments.

### 2.2 Aerosol ionization and mass spectrometry

HP-SPAMS uses a higher frequency diode-pumped Nd: YAG 266 nm (100 Hz, 9 ns pulse width, Centurion Plus, Quantel), focused on the center of the ion source through a UV fused silica plano-convex lens (f=175 mm, SPX026, Newport Corporation). The repetition frequency of the laser was increased 5 times compared to 20 Hz in SPAMS, which could reduce the "busy time" of the laser and improve the temporal resolution. The laser beam homogenization gives the spot approximate

energy density at different locations, thus enhancing the repeatability of the measurement. (Steele et al., 2005) In addition, the laser adopted air-cooling technology, which does not require circulating water to cool the internal laser system and is suitable for vibration environments such as vehicles and shipboard, improving the stable operation.

The detailed design of the bipolar TOF-MS is described in detail in previous works.(Li et al., 2011; Li et al., 2018) Briefly, a wide mass range of ion kinetic energy compensation was produced by an exponential pulse delay electric field so that ions of

the same mass reached the detector simultaneously to achieve an improved resolution. Another advantage of using the pulse



delay extraction technique instead of DC extraction is that it eliminates the problem that charged aerosol particles can deflect in the mass spectrometry electric field conditions, thus improving the instrument's hit rate. (Chen et al., 2020; Clemen et al., 2020) Calibration of each spectrum after the resolution improvement could ensure a mass deviation within ±0.024 Da. (Zhu et al., 2020)

The inhomogeneity of the components of the aerosol particles could result in large differences in the signal response of the microchannel plate detector (1.8 ns single ion response half-peak width), which may reach a maximum of 20 V and a minimum of only a few mV. Therefore, it was not easy to achieve such a high dynamic range acquisition with a single channel using a conventional analog-to-digital conversion (ADC) acquisition card. HP-SPAMS employed a 4-channel data acquisition technique, with positive or negative ion signals being acquired with two channels. As in Figure 2, a single was

split into two similar signals (*S1* and *S2*) with half attenuation through a power splitter (Mini-circuit, ZFRSC-42+). *S1* was acquired by the *CH1* channel (2 mV-500 mV) of the ADC card (Acqiris, U5309-CH4, 8bit, 1GS/s), while *S2* was acquired by the *CH2* channel (20 mV-5 V) after passing through an attenuator (6dB). The data acquired by *CH1* and *CH2* were combined with the algorithm to generate the real ion signal. Therefore, the dynamic acquisition range of 4 mV-20 V using the 4-channel data acquisition method enabled HP-SPAMS to represent the original peaks of the spectrum more accurately on the one hand and to improve the identification of trace components on the other hand. Although it is theoretically possible

to achieve such a dynamic range using a high bit data acquisition card, the SPAMS was usually a single particle single acquisition, which made the ADC acquisition card unable to suppress noise by accumulating acquisitions and thus cannot achieve the theoretical dynamic range in practical applications.

**2.3 Experimental materials and tools**

Standard particle size (100-5000 nm) polystyrene latex microspheres (PSLs), dioctyl sebacate (DOS), sodium chloride (NaCl), and ammonium sulfate (AS) were separately dissolved in an aqueous solution to produce aerosols using a single-jet atomizer (TSI 9302). The resulting aerosol particles were passed through a diffusion dryer to form polydisperse aerosol particles (Figure 3a). To obtain monodisperse particles, particles of different sizes (100-3000 nm) were screened using the aerodynamic aerosol classifier (AAC, Cambustion), and the concentration of particles was measured by a condensation

particle counter (CPC 3775, TSI). Air sampling was performed in the production workshop of Hexin Instruments (Guangzhou Science City, Guangzhou), where daily activities such as instrument commissioning and assembly were performed. Indoor air samples were fed through a carbon black tube into the DUST METER (flow rate 3 L/min, TSI DUSTTRAK II 8530), scanning mobility particle sizer spectrometer (SMPS, flow rate 0.3 L/min, TSI 3321), SPAMS (0.1 L/min, Guangzhou Hexin Instrument Co., Ltd.) and HP-SPAMS (0.3 L/min), respectively (Figure 3b). To compare the

detection capability of the instruments at low concentrations, the fraction of the samples fed by SPAMS and HP-SPAMS was diluted 40 times by an aerosol diluter (Zhanye Dahong Company, equipped with a 3 L/min pump).





## 3 Results and discussion

### 3.1 Improved efficiency and quantity of particle detection

The detection efficiencies of HP-SPAMS and SPAMS for AS, DOS, NaCl, and PSLs particles were compared (Figure 4).
The results show the detection efficiencies of aerosol in the particle size range of 100-3000 nm in both SPMSs, where legend
PMT1 and PMT2 represent the detection efficiencies of the two photomultiplier tubes of SPAMS, respectively. HP-PMT1
and HP-PMT2 are the detection efficiencies of the two photomultiplier tubes of HP-SPAMS detection efficiency. The results
showed that the detection efficiency of HP-SPAMS was higher than that of SPAMS for all the investigated aerosol particles.
Moreover, the HP-SPAMS and SPAMS detection efficiency at PMT1 was higher than that at PMT2 because the particles
underwent dispersive motion after passing through the aerodynamic lens, although the dispersion was very small. Comparing
the samples with different particle morphologies, the detection efficiency of spherical DOS (shape factor $\chi \approx 1$) and PSLs
($\chi=1$) particles was higher than that of non-spherical AS ($\chi \approx 1.1$), and NaCl ($\chi=1.69$–6.27) particles, especially the detection
efficiency of the DOS particle in the range of 300-1000 nm was almost 100 %. (Tavakoli and Olfert, 2014; Zelenyuk et al.,
2006)
Two aspects influenced the enhancement of HP-SPAMS detection efficiency. On the one hand, it was due to the improved
transmission efficiency of the feed system, especially the adoption of a larger critical orifice, resulting in a lower loss rate of
particles before entering the aerodynamic lens (Cahill et al., 2014). On the other hand, the enhancement of the power of the
continuous laser and the reduction of the background noise level improved the ability of the sizing system to detect small
particles and particles at the edge of the Gaussian beam. In addition, HP-SPAMS made it possible to increase the number of
incoming particles by a factor of 3-5 by introducing the APC device, which was determined by the diameter of the critical
orifice. Therefore, the number of particles detected by the HP-SPAMS sizing system could be increased by more than 10
times when considered together.

The hit rate is also an important parameter of the SPMS, representing the ratio of particles capable of measuring the
composition to the number of sizing particles. Figure 5 shows the hit rate curves of HP-SPAMS and SPAMS for different
particle size segments of aerosols in PSLs and workshop air, respectively. Due to the limited number of large particles
collected by SPAMS for air (see later comparing the number of particles in air samples), the hit rate of SPAMS for particles
larger than 1250 nm in size may not be accurate. As seen from Figure 5, the hit rate of HP-SPAMS for PSLs could be
maintained in the range of 200-3000 nm from 80 % to 100 % for PSLs, while the SPAMS was within 20 %, decreasing with
the smaller particle size. This is because the aerosol produced by the atomizer will be charged (Chen et al., 2020), and when
the aerosol is under the action of the direct current (DC) electric field of SPAMS, the particles will be deflected in the
extraction region of the TOF analyzer. On the contrary, HP-SPAMS showed a better hit rate due to the use of DC-extraction
of SPAMS (Chen et al., 2020; Clemen et al., 2020). It could be observed from the particle size segment less than 500 nm that
for SPAMS, the air hit rate was higher than the hit rate of PSLs due to the relatively low charge of particles in the
environment (He et al., 2020). There was a decrease for air particles larger than 1000 nm in both HP-SPAMS and SPAMS




due to the increase of non-spherical particles affecting the focusing of ADLs. Besides the effect of particle trajectory and focus, the hit rate was also influenced by the characteristics of the ablation/ionization laser. Because the components of various particles differed, the particles showed different laser energy absorption cross sections and required different laser energy thresholds to produce ions.(Thomson et al., 1997) Even for the same particle, differences in energy density at the edges and center of the non-uniform Gaussian beam could cause the particles were not ionized.

Further, to compare the instruments' improvement in particle detection quantity, the SMPS, DUST METER, HP-SPAMS, and SPAMS instruments collected and analyzed the air samples simultaneously (Figure 3b). The samples collected by SPAMS and HP-SPAMS were the samples after the diluter (40 times dilution). The variation of particle concentration during 24 h continuous monitoring of air samples is shown in Figure 6. The SMPS monitoring concentration results showed a peak around 12:00 and 16:00, respectively, which may be due to some aerosol particles generated in the workshop, and particle

concentrations continued to decline until the following morning. Pearson correlation analysis between SMPS concentration, PM2.5 concentration, and the data of PMT detection by HP-SPAMS was performed (Table 1). A significant correlation was found between HP-SPAMS measurement data and DUST METER (Pearson correlation>0.97), and this correlation was significantly higher than that of SPAMS results. The correlation coefficient between SMPS and DUST METER and HP-SPAMS was lower (Pearson correlation>0.76) due to the ability of SPMS to measure particles below 100 nm, leading to a

lower correlation of particles due to the influence of small particles.

The statistical analysis of SMPS, SPAMS, and HP-SPAMS particle sizes are detailed in Figure 7. The SMPS is the data after averaging 24-hour sampling points (634 cycles), and HP-SPAMS and SPAMS are the particle size measurement and hit statistical particle distribution, respectively. The peak particle sizes of the three instruments were 88.2 nm, 300 nm, and 380 nm, for SMPS, SPAMS, and HP-SPAMS, respectively. HP-SPAMS was more efficient than SPAMS in measuring particles

in the small particle size range, consistent with the results of the test standard particle (Figure 4). In an environment with PM2.5 concentration of 50 - 200 µg/m3 (Figure 6), and after dilution 40-fold, the number of sizing particles measured by the two instruments was 1,281,846 and 146,600, respectively, indicating that the average particle detection capability was improved about 8.7 times by improving the inlet system and the sizing system. The number of comparisons hit particles was 1002141 (78.2 % hit rate) and 20,943 (14.3 % hit rate), indicating that HP-SPAMS improved the total number of detected

particles by 47.8 times compared to SPAMS. Even in a low PM2.5 concentration environment of 1 µg/m3, the instrument could still detect about 6 particles per second, which is related to atmospheric particles' composition and mixing state. The enhancement of the number of detected particles is very important for low PM2.5 concentrations, improving the temporal resolution of aerosol characterization, such as north or south pole's aerosols, stratospheric cloud condensation nuclear, and coupling with SMPS instruments.

**3.2 Mass resolution and detection sensitivity**

Figure 8 shows the resolution of positive and negative ions for detecting environmental particles for HP-SPAMS and SPAMS. To ensure the accuracy of the resolution calculation, spectrum peaks with peak intensities above 100 mV and





unsaturated (<20 V or <5 V) were selected. It can also be seen from Figure 8 that the resolution of HP-SPAMS using exponential pulse delay extraction is, on average, 2-3 times better than that of the DC-extraction SPAMS, and the average

resolution is up to 2500 at *m/z* 208. The better resolution of HP-SPAMS is due to the exponential pulse delay extraction technique that compensates for the different ion kinetic energy dispersion and solves the problem of the difference in ion generation time due to the laser emission period (several nanoseconds). Of course, not all the peaks of HP-SPAMS have very good resolution due to the various components of the particles.

In addition, Figure 8 shows that the number of HP-SPAMS peaks was significantly higher than that of SPAMS, especially

for *m/z* above 100. It is noteworthy that the ion spectrum peaks with a peak intensity of less than 100 mV were not yet showed in the figure. The increase in the spectral peaks may be due to the following effects: (1) the increase in resolution increased the signal intensity collected by the ADC. (2) it could increase the collision of electrons, ions, and neutral components during the delay time. (Reinard and Johnston, 2008) (3) the particles were better ionized under a uniform laser beam, even if they fell in different areas of the spot. (Steele et al., 2005)

The as-improved resolution could efficiently distinguish between ions with similar mass but different elemental compositions. Table 2 lists the required theoretical resolution at 50 % discrimination of similar ions with *m/z* less than 60 Da. HP-SPAMS could distinguish most of the interferences between metal ions and organic fragment ions, except for $Mg^+$ (23.985) and $C_2^+$ (24.000), since most of the accurate relative atomic masses of most of the metal ions are less than rounded integers. In addition, HP-SPAMS distinguished the interference between some organic fragment ions. Previously, metal ions

were successfully distinguished by metal oxide ions or isotopic peaks (Snyder et al., 2009; Zhou et al., 2020), such as the identification of vanadium-containing particles by $V^+(51)$ and $VO^+(67)$ ions and iron-containing particles by the peak area ratio Fe (56)/Fe (54) >3. These methods can identify the particles containing metals, while some metal-containing particles will be ignored due to the high percentage of organic fragment ions. Figure 9 shows two metal-containing particles, the K-Cu-amine-containing particles (a) and Fe-amine-aged particles (b). In Figure 9a, it can be observed that there are two peaks

of $V^+$ (50.94396) and $C_4H_3^+$ (51.02348) at *m/z* 51, which could indicate that the particle contained metallic vanadium. Likewise, $Fe^+$ (55.93494) and $C_4H_8^+$ (56.06260) could be distinguished in Figure 9b, and the isotopic peaks $Fe^+$ (53.93961) and $Fe^+$ (56.93539) could be identified. Therefore, the improved resolution of HP-SPAMS to better monitor metal-containing particles will help to more accurately investigate the atmospheric chemical processes and pollution involved in metals in atmospheric particles.

**3.3 Analysis of lead-containing particles**

The performance of HP-SPAMS is usually more sensitive and accurate for analyzing aerosol particles, making it possible to obtain valid data for analysis even at low concentrations. For example, the lead-containing particles were compared and analyzed in Figure 7. By screening the particles containing both *m/z* 206 207 208 with not containing *m/z* 202, the number of lead-containing particles detected by HP-SPAMS and SPAMS was obtained as 7955 and 55, respectively. This meant a

difference of about 145 times between each other. The increase in the number of lead-containing particles detected was due





to the increase in the overall number of particles detected by a factor of about 47.8 (1002141/20943) and due to the increase in the detection sensitivity of individual particles. In addition, the 7955 lead-containing particles detected by HP-SPAMS could be further analyzed in time series for changes in the concentration, which is not possible using SPAMS at low concentrations. Thus, the improved instrument performance of HP-SPAMS relative to SPAMS could better characterize

particles, especially in low-concentration environments.

Fifty lead-containing particles detected by SPAMS and HP-SPAMS were randomly selected for each from the results of the above screening, and each particle's *m/z* is shown in Figure 10a and Figure 10b. It could also be observed that the lead-containing particles detected by HP-SPAMS have more spectral peaks, which could be related to the enhancement of the particle ionization and delayed elicitation techniques, and due to the use of high dynamic range data acquisition, which could

acquire a minimum signal of 4 mV. The high dynamic range data acquisition allowed more valid mass spectral peaks, especially for some large *m/z* of organics. The as-improved resolution could completely distinguish adjacent mass spectral peaks, which helps improve the accuracy of isotope ratio measurements. Statistical analysis of the relative peak areas of $^{206}Pb^+$ vs. $^{208}Pb^+$ was performed for all SPAMS and HP-SPAMS-acquired Pb-containing particles (Figs. (c) and (d)). A linear fit to the relative peak areas showed better linearity of the isotopic ratios for HP-SPAMS ($R^2 = 0.92$) due to the better

resolution of HP-SPAMS and the wider dynamic range of data acquisition with peak intensities up to 20 V, which would make it more prominent for the use of isotopic identification, such as source apportionment or mineral identification (Marsden et al., 2018; Souto-Oliveira et al., 2018).

## 4 Conclusions

A new high-performance single-particle aerosol mass spectrometer (HP-SPAMS) was described with an integrated

aerodynamic particle concentration device, dual-polarity exponential pulse delay elicitation, multi-channel data acquisition, and improved aerodynamic lens, sizing system, and ionization system, making the instrument significantly more efficient in quantitative detection of particles. Moreover, the accuracy and sensitivity of quantification of individual particles were also improved.

Comparative analysis between HP-SPAMS and SPAMS showed that the increased inlet flow rate by adding the APC could

enhance the particle number concentration by a factor of 3-5. Combined with the improved aerodynamic lens and sizing system, the transfer and measurement of 100-5000 nm particles are achieved. For spherical particles, the sizing transfer efficiency is almost 70 %~ 100 % in the 300-3000 nm range. A factor of about 5 improved the hit rate by reducing the deflection of charged particle beams through the pulse delay extraction technique. The results of atmospheric aerosols analysis showed that the correlation between the number of particles detected by HP-SPAMS and PM2.5 mass concentration

reached 0.97 (R-square), and its detection efficiency was higher than SPAMS for small particle sizes. The total number of particles effectively detected was improved by approximately 47.8 times.

For the composition detection of individual particles, HP-SPAMS improved the resolution of the method by 2–3 times and could distinguish most of the metal ions and organic fragment ions with more accurate isotopic ratios, enabling this approach



to improve the accuracy of aerosol component identification and aerosol sources. In addition, although the sensitivity of HP-

SPAMS to analyze individual particles cannot be quantified, it can be inferred that HP-SPAMS can improve the sensitivity

toward individual particle detection by improving the number of particles detected (such as lead-containing particles) and the

chance of generation of ions with high mass-to-charge ratio.

**Acknowledgements.** This work was supported by by the National Natural Science Foundation of China (Grant. 41905106).

We would like to thank Professor Zhang Guohua and Professor Hu Ligang for their help in writing this article.

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





**Table 1. Pearson correlation** analysis of SMPS concentrations, PM2.5 concentrations, and the number of particles detected by HP-SPAMS**

|  | SMPS | PM2.5 | HP-PMT1 | HP-PMT2 | HP-PMT1-2 |
|---|---|---|---|---|---|
| SMPS | 1 | 0.818 | 0.815 | 0.786 | 0.768 |
| PM2.5 |  | 1 | 0.984 | 0.982 | 0.977 |
| HP-PMT1 |  |  | 1 | 0.997 | 0.993 |
| HP-PMT2 |  |  |  | 1 | 0.999 |
| HP-PMT1-2 |  |  |  |  | 1 |

**Correlation is significant at the 0.01 level (2-tailed).

**Table 2. Resolution required and achieved for 50 % separation of similar mass number ions with mass numbers less than 60 Da that may be generated by HP-SPAMS**

| Ions | Similar ions | $R^*$ | $R^{**}$ | Ions | Similar ions | $R^*$ | $R^{**}$ |
|---|---|---|---|---|---|---|---|
| $Mg^+$(23.985) | $C_2^+$(24.000) | 2250 | 603±198 | $Ti^+$(47.948) | $C_4^+$(48.000) | 1300 | 1210±273 |
| $Al^+$(26.981) | $C_2H_3^+$(27.023) | 940 | 746±159 | $Cr^+$(49.946) | $C_3N^+/C_4H_2^+$(50.003/50.016) | 1100 | 1321±223 |
| $CHO^+$(29.003) | $C_2H_5^+$(29.039) | 1150 | 829±214 | $V^+$(49.947) | $C_4H_3^+/C_3HN^+$(51.024/51.011) | 920 | 1411±228 |
| $NO^+/CHOH^+$(29.998/30.011) | $CH_4N^+/C_2H_6^+$(30.034/30.05) | 1210 | 755±284 | $Cr^+$(51.941) | $C_4H_4^+$(52.031) | 800 | 1402±282 |
| $P^+$(30.974) | $CH_3O^+$(31.018) | 1010 | 946±232 | $Fe^+$(53.940) | $C_4H_6^+$(54.047) | 790 | 1421±284 |
| $K^+$(38.964) | $C_3H_3^+$(39.024) | 950 | 946±306 | $Mn^+$(54.938) | $C_3H_5N^+/C_4H_7^+$(55.042/55.055) | 770 | 1368±355 |
| $Ca^+$(39.964) | $C_3H_4^+$(40.031) | 850 | 1018±314 | $Fe^+$(55.935) | $C_4H_8^+$(56.063) | 630 | 1435±299 |
| $K^+$(40.962) | $C_3H_5^+$(41.039) | 820 | 1081±298 | $CaOH^+$(56.965) | $C_4H_9^+/C_3H_7N^+$(57.070/57.058) | 780 | 1300±356 |
| $CHNO^+/CH_3CO^+$(43.006/43.018) | $C_3H_7^+$(43.055) | 1280 | 935±263 | $Ni^+$(57.935) | $C_3H_8N^+/C_4H_{10}^+$(58.066/58.078) | 700 | 1579±203 |
| $SiO^+/Ca^+$(43.972/43.955) | $C_2H_6N^+$(39.963) | 800 | 1242±226 | $C_5^+$(60.000) | $C_3H_{10}N^+$(60.081) | 1050 | 1444±420 |

* Resolution required for 50 % separation of two similar ion peaks.

** Resolution statistics of HP-SPAMS at different mass-to-charge ratio.



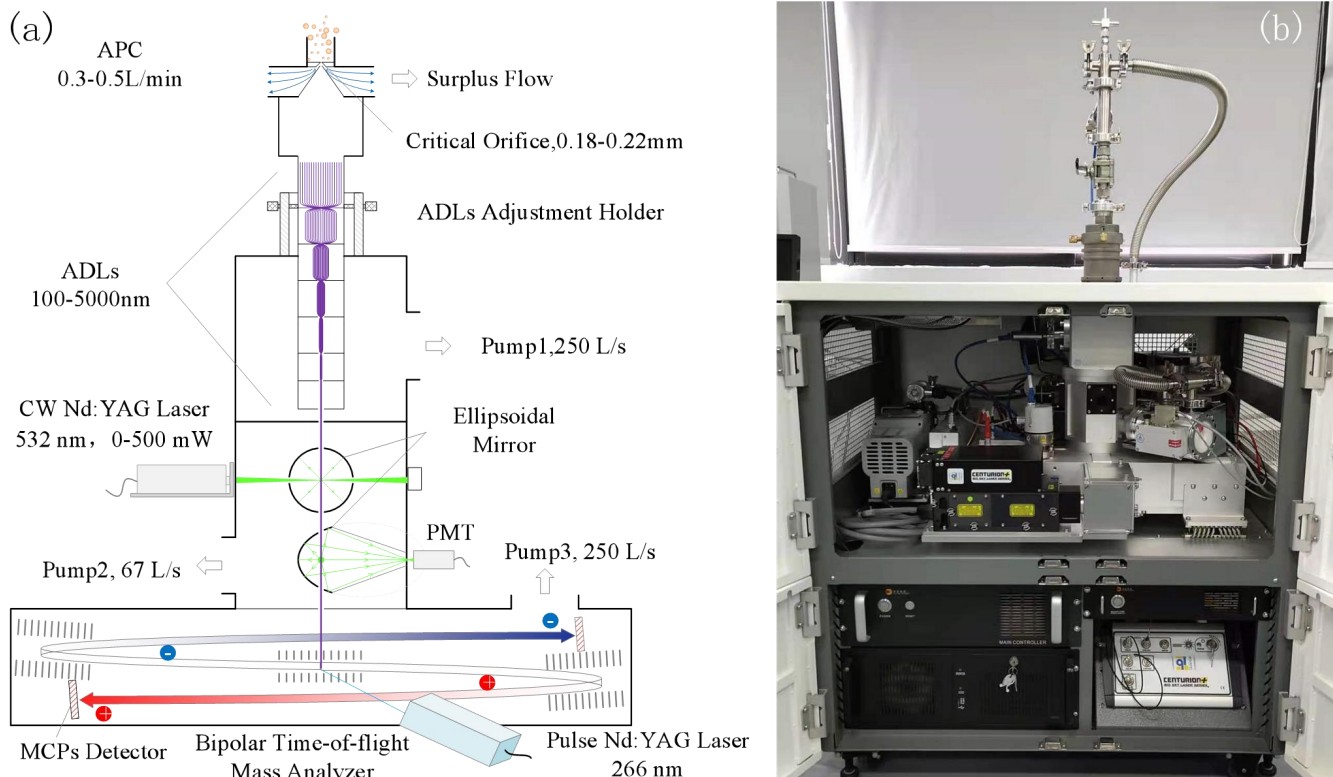

**Figure 1: Schematic diagram and photograph of the HP-SPAMS instrument. (a) The aerosol particles were introduced through an APC into an improved aerodynamic lens (ADLs) to focus into a particle beam. A scattering system measured the particle size with two higher power continuous lasers (532nm). Particle chemical components were obtained by bipolar time-of-flight mass spectrometry analysis of ions derived from a pulsed laser (266nm) ablation/ionization of particles. (b) The instrument's main body was placed in the upper part of the frame, and the lower part was composed of the power supply, control unit, and pump.**





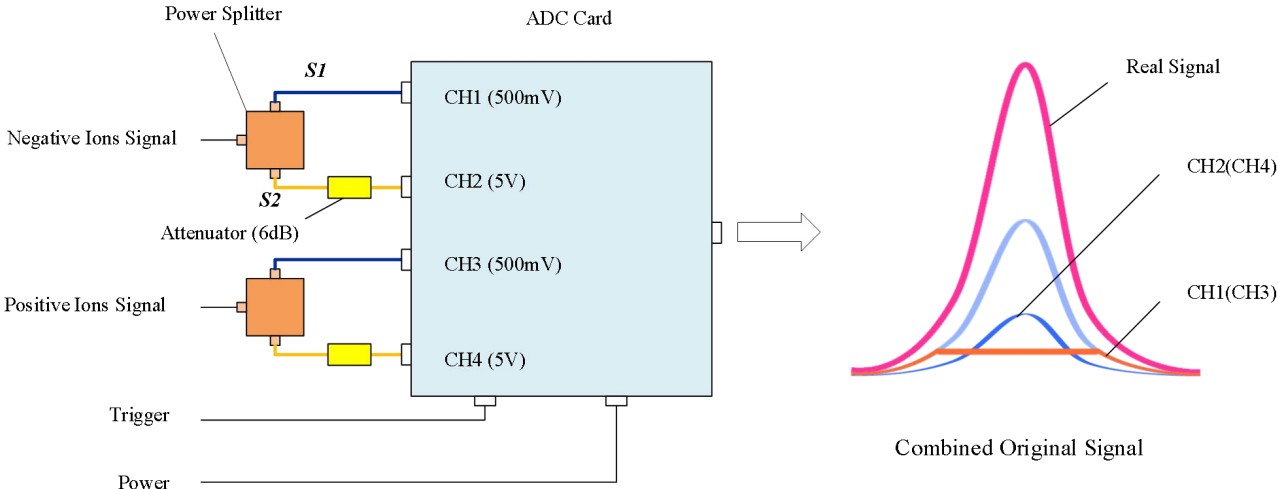

**Figure 2: Schematic diagram of the working principle of multi-channel data acquisition. The ion signals were acquired separately using two channels with different ranges (500mV and 5V) of the ADC (8-bit). Finally, the real signal intensity was obtained by synthesizing the signals of two channels through software algorithms, and the dynamic acquisition range reached 4mV~20V.**


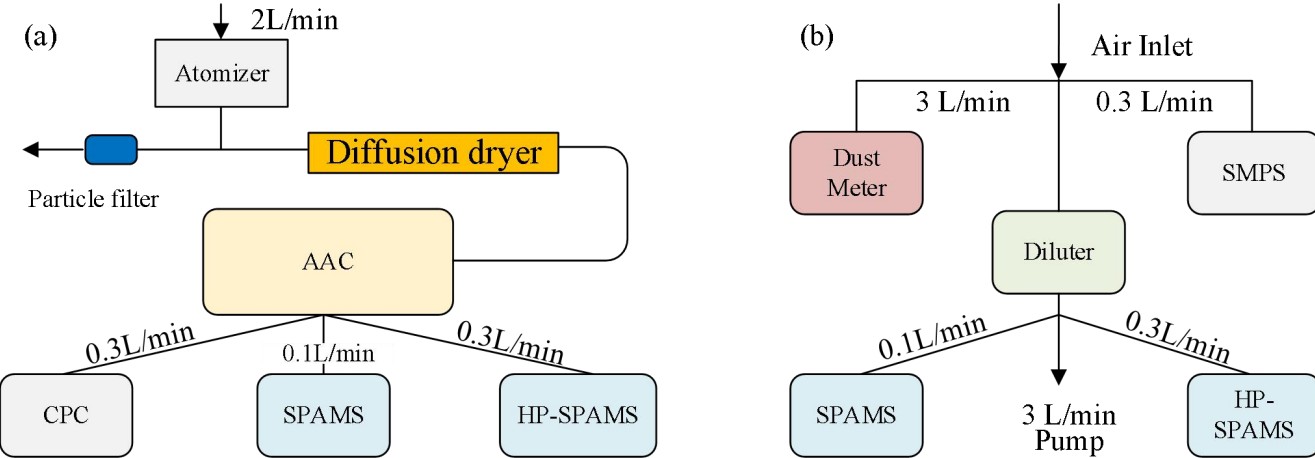

**Figure 3: Standard sample and laboratory air experiment diagram. (a) The atomizer produced standard particles (100-5000 nm) and dried them with a diffusion dryer. The monodisperse particles (100-3000 nm) were screened using the aerodynamic aerosol classifier (AAC, Cambustion). Moreover, the monodisperse particles are measured by a condensation particle counter (CPC 3775, TSI), SPAMS, and HP-SPAMS. (b) Air samples were fed through a carbon black tube into the dust meter, SMPS, SPAMS, and HP-SPAMS, respectively. To further compare the detection capability of the instruments at low concentrations, the fraction of the samples fed by SPAMS and HP-SPAMS was diluted 40 times by an aerosol diluter.**




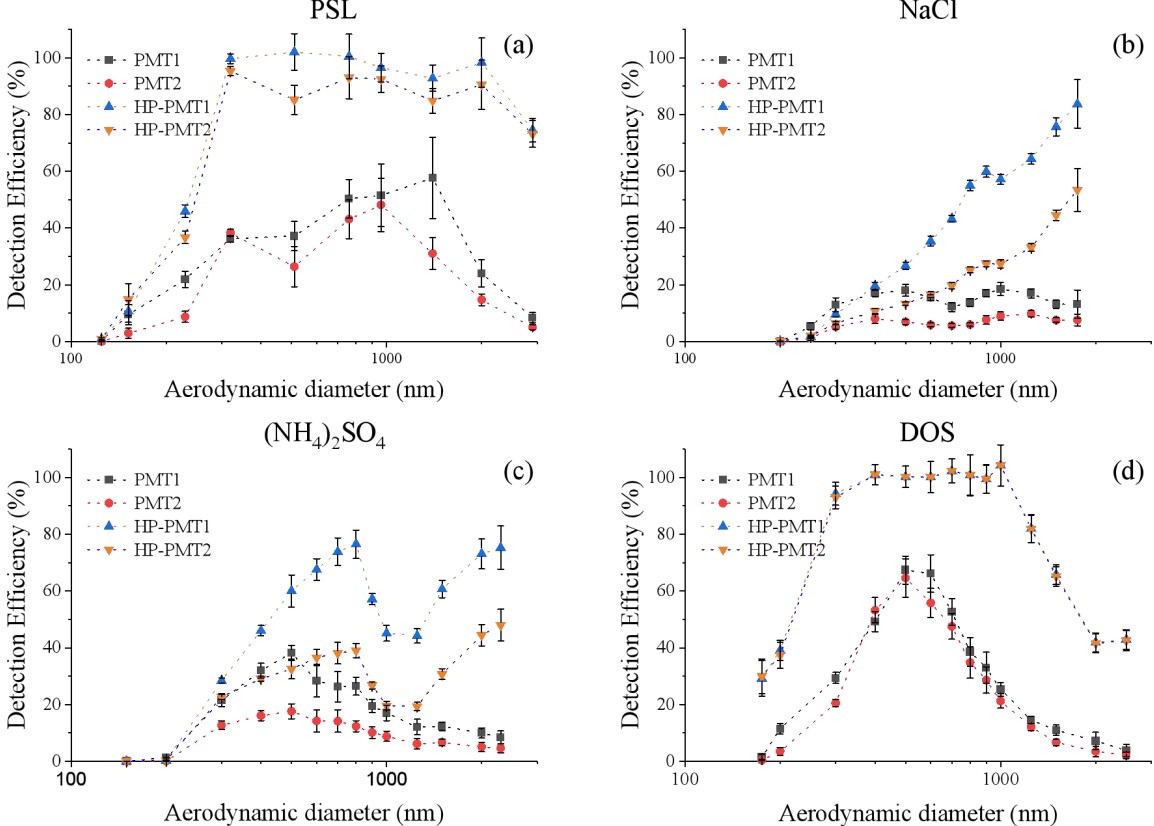

**Figure 4: Comparison of HP-SPAMS and SPAMS for different particle sizing detection efficiency ranged from 100~3000 nm. The**
**PMT1 and PMT2 represent the detection efficiencies of the two photomultiplier tubes of SPAMS, respectively, and HP-PMT1 and**
**HP-PMT2 are the detection efficiencies of the two photomultiplier tubes of HP-SPAMS detection efficiency.**



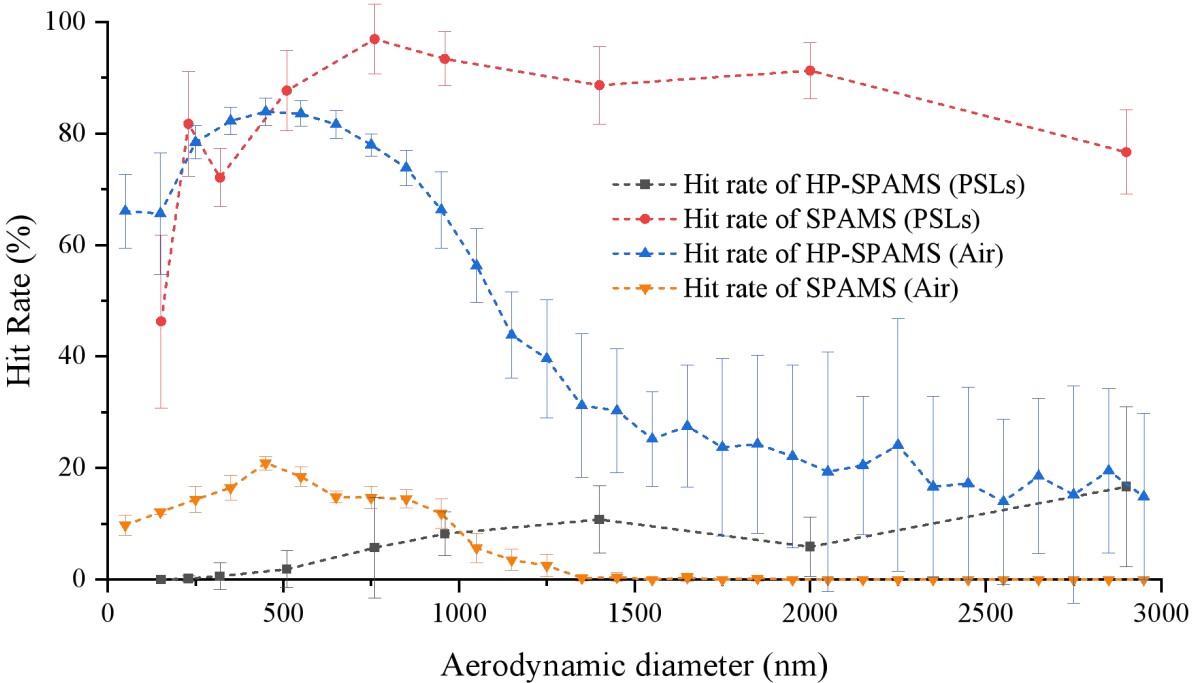

**Figure 5: The HP-SPAMS and SPMAS hit rate plots for PSLs and air at different particle sizes. Both for measuring PSLs and air, HP-SPAMS had a higher hit rate than SPAMS. For PSLs particles, the hit rate of HP-SPAMS could be maintained in the range of 200-3000 nm from 80 % to 100 % for PSLs, while the SPAMS was within 20 %.**



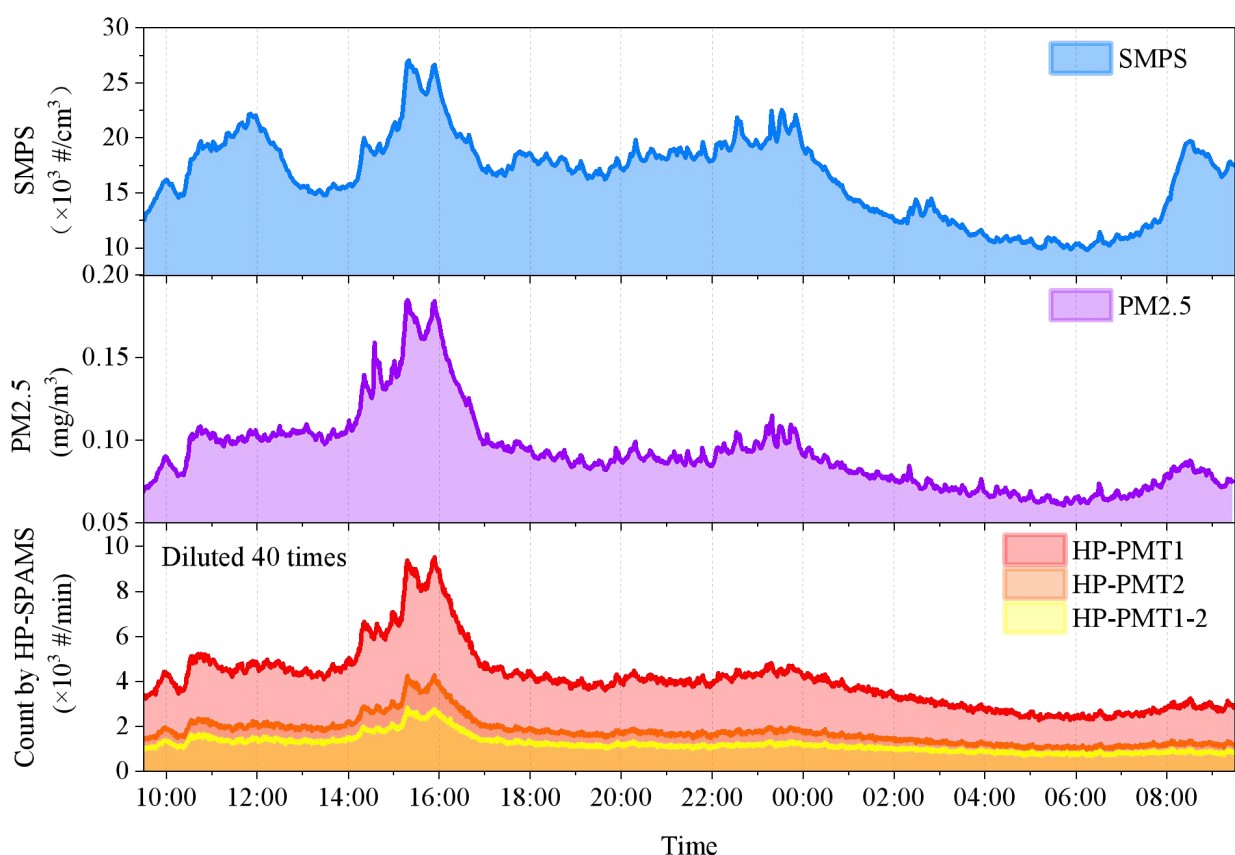

**Figure 6: The SMPS, DUST METER, HP-SPAMS instrument air collection number concentration variation curve.**



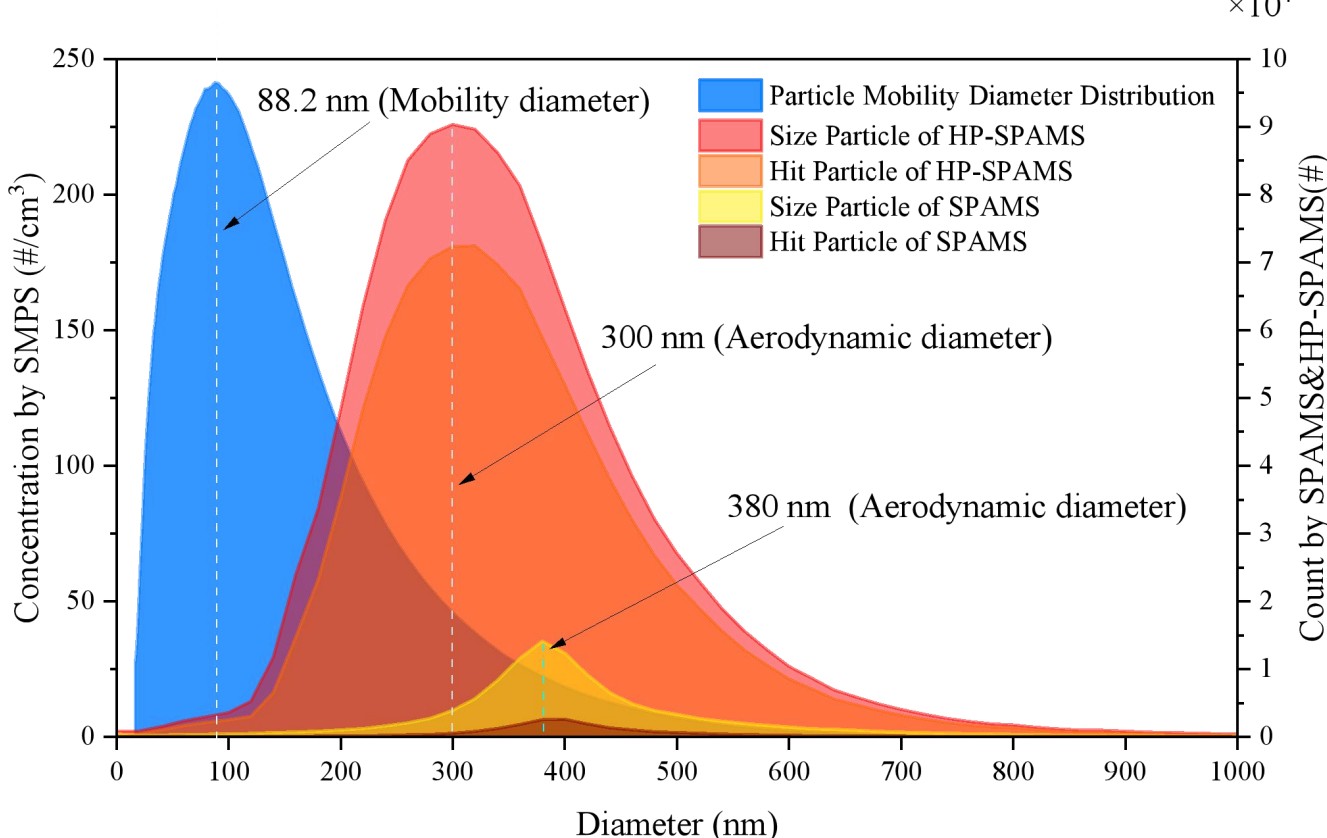


**Figure 7: Particle size distribution of SPMS, HP-SPAMS and SPAMS detection. The peak particle sizes of the three instruments were 88.2 nm, 300 nm, and 380 nm, respectively. The number of sizing particles measured by the two instruments was 1,281,846 and 146,600, respectively, indicating that the average particle detection capability has improved about 8.7 times by improving the inlet and sizing systems. The number of comparisons hit particles was 1002141 (78.2 % hit rate) and 20,943 (14.3 % hit rate),**

**which indicated that HP-SPAMS improved the total number of detected particles by 47.8 times over SPAMS.**



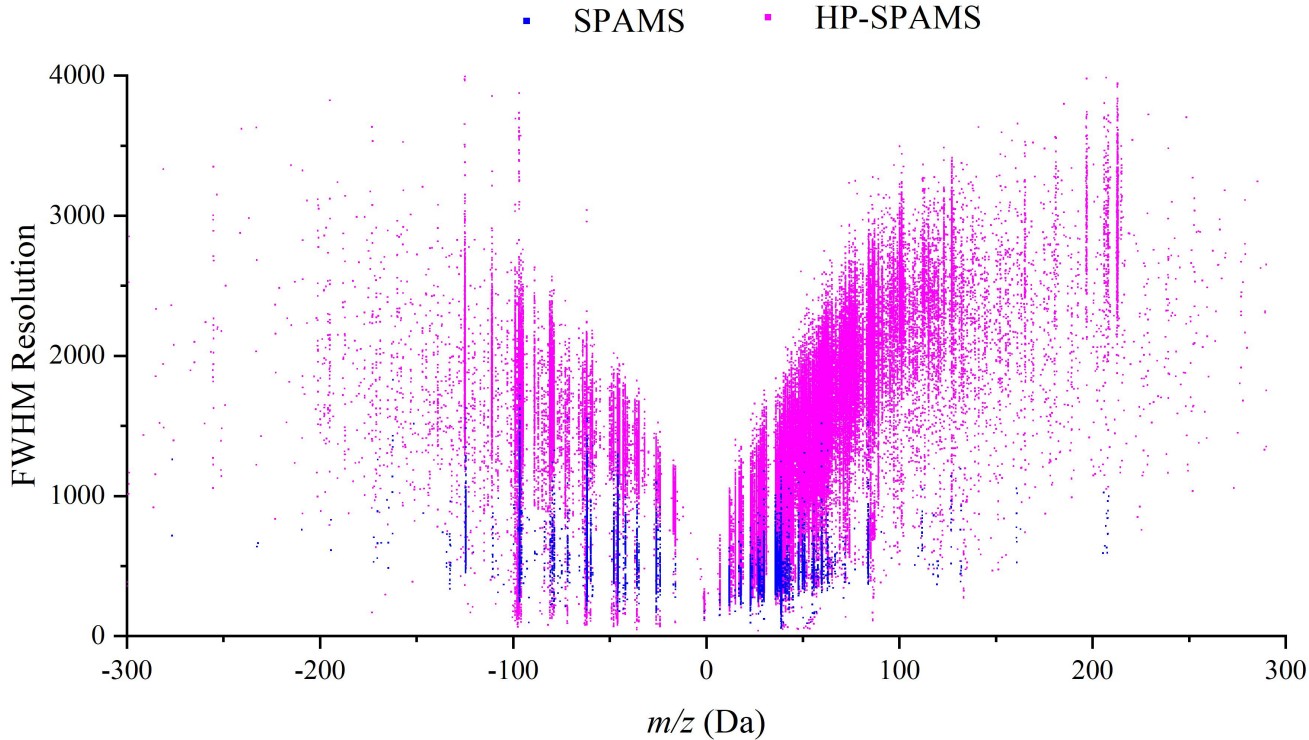

**Figure 8: HP-SPAMS and SPAMS positive and negative ion resolution scatter plot.**





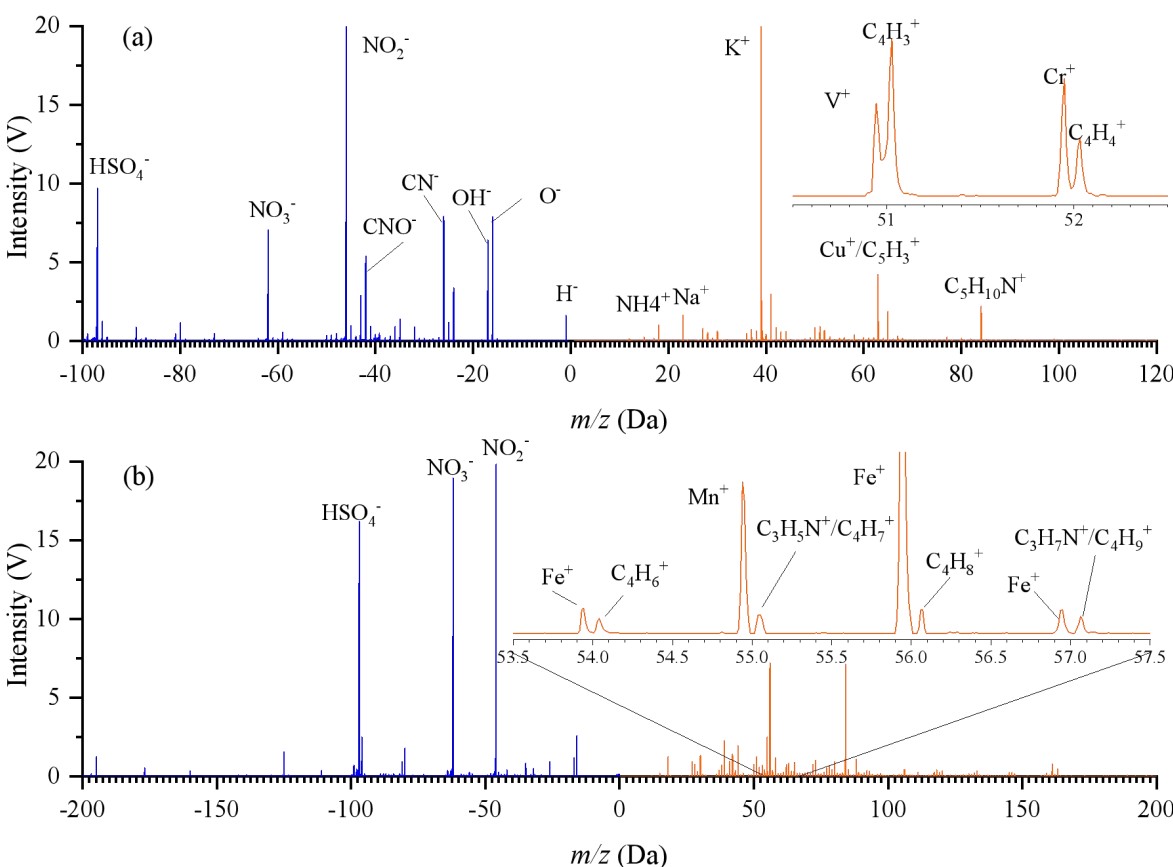

**Figure 9: Example particles of high-resolution spectra (a) K-Cu-amine containing particles (b) Fe-amine aged particles.**






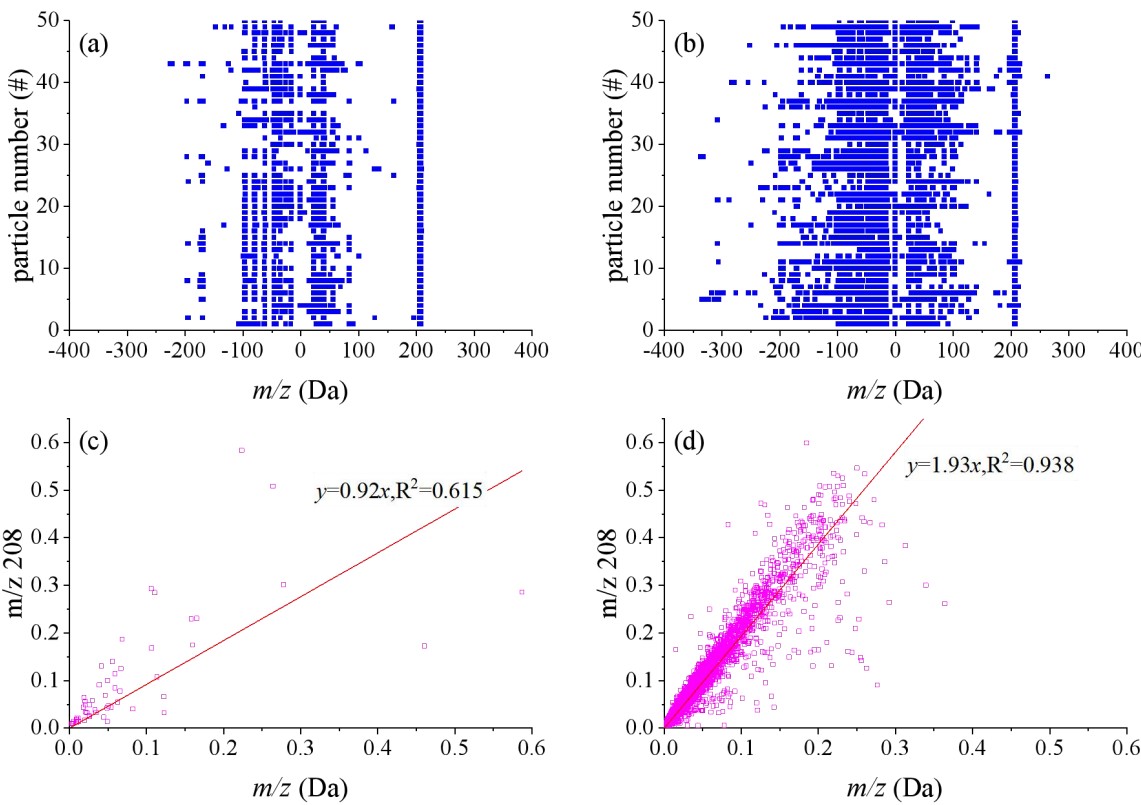

**Figure 10: (a) and (b) show the scatter plots of the mass-to-charge ratios of the ions of the lead-containing particles from SPAMS and HP-SPAMS, respectively. (c) and (d) showed the scatter plots of the relative peak areas of $^{206}$Pb$^+$ vs. $^{208}$Pb$^+$ for the lead-containing particles from SPAMS and HP-SPAMS, respectively.**