# Peer review of "Development and characterization of a High-Performance Single-Particle Aerosol Mass Spectrometer (HP-SPAMS)"

_EGUsphere, 2022_

## Author Response (AR1)

**Responds to the Referees**

**Referee #1**

The manuscript by Du et al. entitled 'The design and characterization of a High-Performance Single-Particle Aerosol Mass Spectrometer (HP-SPAMS)' describes the development and improvement of single particle mass spectrometry in the inlet system, higher detection efficiency and hit rate, and greatly improved mass resolution in the analyzer. The results shown here impressively demonstrate the possibilities of this technology and clearly show advantages and actual scientific questions, which will be addressed here.

However, questions remain and should be discussed and improved before publication. Overall, this manuscript should be accepted for publication after improvements focused on the specific comments below.

More general comments:

- Add spot sizes for ionization and sizing lasers.
**Answer:** Thank you for your suggestion, the description of the laser spot size " Two beams of continuous-wave Nd:YAG (532 nm) laser (500 mW, LaserWave, LWGL532 nm-500 mW) spaced orthogonally 6 cm apart were focused by a plano-convex lens (PLCX-25.4-51.5-C-532, CVI Laser Optics) into a 300 μm-sized spot at the first focus of the ellipsoidal mirror, respectively. Scattered light is produced as the particles pass through the laser beam and is focused by the ellipsoidal mirror to the photomultiplier tube (PMT, H10721-110, Hamamatsu) for detection. " and " HP-SPAMS uses a diode-pumped Nd: YAG 266 nm (100 Hz, 9 ns pulse width, Centurion Plus, Quantel) that is focused by a UV fused silica plano-convex lens (f=175 mm, SPX026, Newport Corporation) into a 300 μm square uniform spot at the center of the ion source " has been added.

- Add energies of ionization lasers of SPAMS and HP-SPAMS because it highly influences the hitrate.
**Answer:** Thanks, both HP-SPAMS and SPAMS operate at laser energies of about 0.5 mJ/pulse during the experiments, and it has been added in Section 2.3.

- Define 'hitrate' (' of mass specs in relation to # at PMT2 ??)
**Answer:** I'm sorry if the definition of hit rate confused you. The hit rate is defined as the ratio of the number of particles with mass spectra to the number of sizing particles, where the number of sizing particles refers to the number of particles accurately measured by the sizing system at the time the ionizing laser can be emission.

- The detection efficiency is given in % -> in relation to what? How is the higher repetition rate of the newer system, and its timing electronics taken into account?
**Answer:** (1) Sorry for not defining this detection efficiency well, it may be more acceptable to use the scattering efficiency. The scattering efficiency of particle is defined as the ratio of the response

pulse count of the photomultiplier tube to the total number of injected particles, in which the total number of injected particles can be obtained from the particle concentration and the injection flow rate. Taking the 960 nm PSL as an example, the number of particles detected by CPC is shown in Figure S1, and the average number concentration is 8.8 particles/cm³. For HP-SPAMS, the average number of particles detected by the two laser beams is 64.9 (particles/s) and (62.2 particles/s), and the scattering efficiency is 96.3% and 92.3%, respectively, according to the following equation. Therefore, the scattering efficiency is calculated with the normalization of the inlet flow already.

$$\text{Scattering efficiency} = \frac{response\ pulse\ by\ instrument}{CPC\ concentration \times inlet\ flow \times time} \qquad (\text{equation 1})$$

[Figure]

Figure S1: Concentration of 960 nm PSL for CPC detection and pulse counting for HP-SPAMS.

(2) The maximum frequency is the laser output frequency of 100Hz, which has been judged logically on the timing card and can meet the maximum 500/s trigger.

(3) The new timing card uses a serial port (baud rate 115200) which can transmit 11520 bytes a second and does not transmit the complete waveform of the PMT in the data transmission. On the one hand, only the pulse counts of individual PMTs are counted, and on the other hand, only the time-of-flight information (4 bytes) of the measured particles is transmitted to the host computer after judging the exact particles by a certain logic. Therefore, it is possible to meet the requirements.

- Figure 4: PMT2 is of course always lower in signal number and therefore detection efficiency, why is the detection efficiency in (d) with all errorbars the same as for PMT1? This should at least show a difference due to the divergence of the particle beam.

Answer: Thanks for your reminding. I think this may be explained by: (1) for dioctyl sebacate (DOS, density 0.90 g/cm³, shape factor = 1), the divergence of the lens is smaller for spherical

particles; (2) the density of the DOS particles used in Figure 4d is 0.9 g/cm3, which is smaller than the standard PSL (1.05 g/cm3). Therefore, the geometry of the particles with the same aerodynamic diameter will be larger ( $\boldsymbol{d_{va} = d_p \frac{\rho_p}{\chi \rho_0}}$, *see Alla Zelenyuk* **2011** ) and the intensity of scattered light will be enhanced, which may be the reason why the errorbar of HP-PMT1 and HP-PMT2 of DOS are similar. Especially for DOS particles of 400-1000 nm particle size, the pulse counts of HP-PMT1 and HP-PMT2 are almost the same, so the errorbar of this particle size is almost the same.

[Figure]

Figure S2: Scattering efficiency of HP-SPAMS and SPAMS for DOS

- Figure 4: Please add data from CPC for better understanding of the detection efficincy and normalize to to sample volume, so that the two systems can be better compared.

**Answer:** Thank you for your suggestions. Here the detection efficiency (has been changed to scattering efficiency) is already normalized, see equation 1. We have used the measured concentrations of CPC in the calculation of the efficiency and normalized them. And the method of calculation we have also added in the manuscript.

- Figure 6: please also add a plot as the third one for the (non-HP) SPAMS.

**Answer:** We are very sorry that (non-HP) SPAMS uses an old timing card, which does not have the function to record the pulse counts of photomultiplier tubes, so we cannot show the data of this part. Although the SPAMS data are missing, this does not affect the analysis of the correlation between the scattering counts of HP-SPAMS and PM2.5. As a supplement, the number of sizing particles and hitting particles are recorded for SPMAS and HP-SPAMS (Figure S3).

[Figure]

Figure S3: The number of sizing particles and hitting particles of SPMAS and HP-SPAMS. Size particle is the particle accurately measured by the sizing system and emitted by the ionizing laser. Hit particle is the number of particles with a spectrum after laser emission. Mass particle is a particle that is present in both positive and negative mass spectra.

- Figure 7: right side y-axis update to #/time (24 h ??)
- Figure 7: normalize to sample flow (ml/ccm^3 -> like SMPS)

**Answer:**(1) Thank you for your suggestion, we have modified it.

(2) Here we have normalized to show the average concentrations detected by both instruments under 24h hourly collection as shown in Figure S4. As can be seen from the figure, the difference in the number concentrations detected by HP-SPAMS and SPAMS is reduced after normalizing the flow rate, but HP-SPAMS still has an enhancement in the whole particle size range, especially in the small particle size range.

[Figure]

Figure S4: Particle size distribution of SPMS, HP-SPAMS and SPAMS detection during 24 hours. The peak particle sizes of the three instruments were 88.2 nm, 300 nm, and 380 nm, respectively.

The number of sizing particles measured by the two instruments was 1,281,846 and 146,600, respectively. The number of comparisons hit particles was 1002141 (78.2 % hit rate) and 20,943 (14.3 % hit rate).

Line 95: Why is this size used? Is there the possibility to use also even larger sized orifice with higher flows and even more particles? Is the flow through the aerodynamic lens also higher or is it the same as in the older version?
**Answer:** (1) With a size in the range of 0.18-0.22 mm, combined with a virtual concentration device, it is possible to increase the total inlet flow rate of the instrument and to ensure that the pressure at the front of the aerodynamic lens remains in the range of 1.8-2.2 Torr ;(2) Using a higher pumping speed and a larger bore, more particles can be obtained, but not endlessly, as it will be limited by the sizing system; (3) The aperture of the aerodynamic lens has been improved, but the flow through is still about 100mL/min (at standard conditions).

Line 176: Not in every case the number of particles is 10 times as high, for NaCl particles the difference is sometimes only about one third. Furthermore, as shown in Figure 4, the difference is size and particle type dependent. Please revise formulation more precisely and elaborate.
**Answer:** Thank you for correcting our mistake, where it is true that not all particles meet the 10 times efficiency (see Figure S5). As can be seen from the graph, the number of particles detected differs by a factor of 1-40 for different particle size ranges. And the multiplier increase is different for different types of particles. Here we modify it to "the number of particles detected by the HP-SPAMS sizing system could be greatly increased, and it is related to the particle size and the type of the particles."

[Figure]

[Figure]

Figure S5: The ratio of pulse counts detected by HP-SPAMS and SPAMS in the simultaneous injection condition.

Line 183: Compared to what number? PMT2 or CPC? Please add information on count number of CPC 1/ccm^3

**Answer:** Here it may refer to line 181. We intended the comparison to be between the particle size distributions measured by HP-SPAMS and SPAMS to show that SPAMS has a small number of acquisitions in the large particle size segment and is not statistically significant. Here we has deleted the " (see later comparing the number of particles in air samples)" to avoid ambiguity.

Line 183/184: Both are decreasing with smaller particle sizes, please revise formulation more precisely and elaborate.

**Answer:** Thank you for your suggestion, here is modified to "As seen from Figure 5, the hit rate of HP-SPAMS for PSLs could be maintained in the range of 200-3000 nm from 80 % to 100 % for PSLs, while the SPAMS was within 20 %. In the range of 100 to 500 nm, the hit rate of both instruments decreases with smaller particle size".

Line 209: Should be: ...SMPS, HP-SPAMS, and SPAMS respectivley... change word order

**Answer:** Thanks. It has been corrected from "The peak particle sizes of the three instruments were 88.2 nm, 300 nm, and 380 nm, for SMPS, SPAMS, and HP-SPAMS, respectively" to "The peak particle sizes of the three instruments were 88.2 nm, 300 nm, and 380 nm, for SMPS, HP-SPAMS, and SPAMS, respectively".

Line 210: This is a total value not taking into account the sample flow rate, please add normalized values.

**Answer:** Thank you for your suggestion, we added a normalized average data. Here is modified to "…the number of sizing particles measured by the two instruments was 1,281,846 (2.8 particle per milliliter on average) and 146,600 (1.02 particle per milliliter on average)…"

All in all, the first paragraph in the results section (lines 159 - 194) is very confusing and should be fundamentally revised. The comparisons with the previous and older measurement system are

misleading in this form, since no laser energies and spot sizes are given and the results have not been normalized. This will make the difference between the systems considerably smaller, nevertheless this is a very good and useful improvement of the technology.

**Answer:** Thank you very much for your suggestions. We have made changes to this, in summary: (1) redefining the detection (scattering) efficiency and hit rate to make these concepts and expressions clearer. (2) Normalized the presentation of these data, such as scattering efficiency, particle concentration, etc., to allow better comparison between instruments. (3) Some descriptive errors are corrected.

**Referee #2**

Review of Du et al., The design and characterization of a high-performance single-particle aerosol mass spectrometer (HP-SPAMS)

This manuscript is, as the title implies, a technical description of a single particle mass spectrometer. This is a commercial instrument, but the description is primarily technical and does not feel too much like an advertisement. It falls reasonably into the type of paper that describes the performance of a commercial instrument.

The technical specifications are in some respects quite impressive. This instrument is, for example, significantly better than the ATOFMS sold some years ago in the US by TSI, Inc. The writing is good and the references are balanced. The manuscript should be publishable with minor revisions.

1) One general change that would benefit the manuscript is to do fewer comparisons to the previous SPAMS instrument developed by the same group and emphasize absolute performance more. The authors are justifiably proud of their improvements, but the eventual paper should be presented for a general audience of all users of single particle mass spectrometers, not just those who own an older instrument developed/sold by the same authors. An example of something to de-emphasize or delete: "the number of lead-containing particles is 145 times higher than that detected by SPAMS" (abstract). An example of something to emphasize more: the comparison in Figure 4 of the detection efficiency of various kinds of particles, presumable due to spherical or non-spherical shapes. This is a very useful comparison between types of particles. These are just examples: there are probably dozens of places in the manuscript where the comparisons to the older SPAMS could be reduced. A few are OK, but only a few.

**Answer:** Thank you for your suggestion. We have also noticed this problem and have added comparative descriptions with ATOFMS, LAAPTOF, as well as reduced some comparative descriptions with older instruments, in order to better demonstrate the absolute performance of the instruments. In detail, we have made the following changes to the manuscript:

(1) Revise the abstract section "For the analysis of individual particles, the HP-SPAMS's improved resolution helps distinguish between most organic fragment ions and metal ions and facilitates the analysis of complex aerosol particles" to " For the analysis of individual particles, HP-SPAMS achieves an average mass spectral resolution of 2500 at m/z 208, which helps distinguish between most organic fragment ions and metal ions and facilitates the analysis of complex aerosol particles."

Delete " for example, HP-SPAMS can completely differentiate the isotopes of lead elements and

the number of lead-containing particles is 145 times higher than that detected by SPAMS. " In the abstract.

(2) Delete " The repetition frequency of the laser was increased 5 times compared to 20 Hz in SPAMS, which could reduce the "busy time" of the laser and improve the temporal resolution. " in Section 2.2.

(3) Add "This is due to the fact that for irregular particles there is a force perpendicular to the direction of gas motion, while for spherical particles this force is zero, resulting in greater dispersion of non-spherical particles after passing through the lens system. (Liu et al., 1995)" in section 3.1.

(4) Delete "… and this correlation was significantly higher than that of SPAMS results." in section 3.1.

(5) Delete" … indicating that HP-SPAMS improved the total number of detected particles by 47.8 times compared to SPAMS."

(6) Added description of resolution compared to other instruments (LAAPTOF and A-ATOFMS). Change "It can also be seen from Figure 8 that the resolution of HP-SPAMS using exponential pulse delay extraction is, on average, 2-3 times better than that of the DC-extraction SPAMS, and the average resolution is up to 2500 (Full Width at Half Maximum) at m/z 208. This resolution is better than that of instruments of the same type such as LAAPTOF (600) and A-ATOFMS (1072)." to " It can also be seen from Figure 8 that the resolution of HP-SPAMS using exponential pulse delay extraction is, on average, 2-3 times better than that of the DC-extraction SPAMS, and the average resolution is up to 2500 (Full Width at Half Maximum) at m/z 208. This resolution is better than that of instruments of the same type such as LAAPTOF (600) and A-ATOFMS (1072)" in Section 3.2.

(7) Delete" This meant a difference of about 145 times between each other." and" In addition, the 7955 lead-containing particles detected by HP-SPAMS could be further analyzed in time series for changes in the concentration, which is not possible using SPAMS at low concentrations. Thus, the improved instrument performance of HP-SPAMS relative to SPAMS could better characterize particles, especially in low-concentration environments" in Section 3.2.

2) One important technical detail that is missing is a list of the spot sizes of the detection and ionization lasers. One cannot interpret the performance of the aerodynamic lens without knowing how big of a target is provided by the detection lasers beams. And one needs the ionization laser spot size to know the fluence available to ionize particles.

**Answer:** Thank you for your suggestion, we have added the spot size in Section 2.1 and Section 2.2. " Two beams of continuous-wave Nd:YAG (532 nm) laser (500 mW, LaserWave, LWGL532 nm-500 mW) spaced orthogonally 6 cm apart were focused by a plano-convex lens (PLCX-25.4-51.5-C-532, CVI Laser Optics) into a 300 μm-sized spot at the first focus of the ellipsoidal mirror, respectively. Scattered light is produced as the particles pass through the laser beam and is focused by the ellipsoidal mirror to the photomultiplier tube (PMT, H10721-110, Hamamatsu) for detection. " and " HP-SPAMS uses a diode-pumped Nd: YAG 266 nm (100 Hz, 9 ns pulse width, Centurion Plus, Quantel) that is focused by a UV fused silica plano-convex lens (f=175 mm, SPX026, Newport Corporation) into a 300 μm square uniform spot at the center of the ion source. "

3) I found the discussion of the aerodynamic concentrator incorporated into the aerodynamic focusing inlet to be confusing. Maybe because it is incorporated into the inlet, it was never clear to me what the baseline was for its performance. In the abstract it says that it "enables concentration… but a factor of 3-5 times". Compared to what? And then in section 3.1 I was not clear in the discussion of detection efficiency what the maximum detection efficiency should be. Should it be 100%? Or perhaps the maximum should be 300 to 500%, because the best possible performance would be 100%, but the concentrator would then multiply that by 3 to 5? I think a clear definition of how the baseline is defined would solve these. By the way, incorporating the aerosol concentrator into the inlet instead of having a separate concentrator upstream looks like a clever idea with several advantages.

**Answer:** Thank you very much for your suggestion, we have made the following changes to this:

(1)Change "enables concentration… but a factor of 3-5 times" to "The combination of an aerodynamic particle concentrator (APC) system and a wide range of aerodynamic lenses (ADLs) enables the concentration of particles in the 100-5000 nm range. Using APC increases the instrument inlet flow by a factor of 3-5".

(2)We have changed "detection efficiency" to "scattering efficiency". The scattering efficiency of particle is defined as the ratio of the response pulse count of the photomultiplier tube to the total number of injected particles, in which the total number of injected particles can be obtained from the particle concentration (from CPC) and the injection flow rate. Therefore, the scattering efficiency is the result obtained after normalizing the inlet flow and the maximum scattering efficiency value is 100%.

4) It would be interesting to see a few spectra on a logarithmic scale, perhaps in supplemental material, to see the dynamic range from the high range/low range digitization and whether or not that induces any artifacts.

**Answer:** Thanks to your suggestion, we have shown the example data in logarithmic scale in the supplemental material Figure S6. It is worth noting that since there is a zero value in the original intensity, therefore, the intensity is overall biased by 0. 1 mV when plotting this graph. From Fig. S6(a) and Fig. S6(b), it can be seen that the maximum data acquisition range can reach 20V (K+), and the smallest signal can be acquired to 0.004V. Fig. S6(c) shows that this is a standard EC containing particle, and from Fig. S(d), which is displayed in logarithmic scale, it can be seen that this EC particle still contains a very small amount of OC.

[Figure]

Figure S6: Example particles of mass spectra in linear scale and logarithmic scale (a)&b Na-K containing particles; (c)&(d) EC containing particle.

5) Near line 104, the pressure would be helpful as well as the orifice diameters. I think the symbol/line labels in Figure 5 may be incorrect.

**Answer:** Thank you for your suggestions:

(1) We have added "The aerodynamic inlet pressure is in the range of 2.0 ± 0.2 Torr and the exit nozzle size is 3 mm." in Section 2.1.

(2) we have exchanged the "Hit rate of HP-SPAMS (PSLs)" with "Hit rate of SPAMS(PSLs)" in legend.

---

## Author Response (AR3)

**Responds to the Referees**

**Referee #4**

This manuscript describes the comparison of the performance between the "High-Performance" HP-SPAMS and the prior version (SPAMS). While limited design details are provided, data is included for the scattering efficiency and hit fraction vs size for both instruments, as well as mass spectral resolution. Yet, there are many general statements in the manuscript for which details/data are not provided, or for which the authors cite a manuscript for a different single-particle mass spectrometer. Line numbers referred to in this review correspond to the track changes version of the manuscript.

1. As noted in the previous reviews, details are missing from the Section 2 description of the HP-SPAMS. Detailed descriptions of the specific parts that we changed (including both the previous and current parts) are needed. Some details were added in the revision, and this is helpful. Yet, it still is often not clear what exactly changed between the two versions of the instrument, and most of the statements are vague/general. The improvements made to the instrument appear to include adding an aerodynamic particle concentrator system, applying delayed ion extraction, using a high frequency YAG laser, and adding four-channel data acquisition.

**Answer:** Thanks to your comments, we have described the instrument improvement points in more detail. In addition to improved aerodynamic particle concentrators, delayed ion extraction, high frequency YAG laser, and adding four-channel data acquisition, as well as including new aerodynamic lenses for 100-5000 nm particle, high-power sizing lasers. These techniques are described in detail or cited previous article in the manuscript Section 2.1 & 2.2. For example, we added "Compared to SPAMS, the laser power is increased from 75 mW to a maximum of 500 mW, which enhances the intensity of the light scattered by the particles." in line 125. "Compared to the ionized laser of SPAMS, the laser of HP-SPAMS contains two features. One is that the laser frequency has been increased from 20 Hz in SPAMS to 100 Hz. The second is that SPAMS employs a laser with a Gaussian beam, while HP-SPAMS employs a laser with a homogenized spot." in line 140.

2. The authors have previously published focused manuscripts about the design and characterization of the two main improvements to instrument:

1) Delayed ion extraction:
- Li et al. 2018, J. Am. Soc. Mass Spec., "Improvement in the Mass Resolution of Single Particle Mass Spectrometry using Delayed Ion Extraction"
- Chen et al. 2020, Atmos. Meas. Tech., "Increase of the particle hit rate in a laser single-particle mass spectrometer by pulse delayed extraction technology"

2) Improved aerosol inlet:
- Du et al. 2023, Atmosphere, "Design and simulation of aerosol inlet system for particulate matter with a wide size range"

While Li et al. (2018) and Chen et al. (2020) are cited, it is not clear that that details of the design

and previous characterization can be found in those manuscripts, and it would be very helpful for the current manuscript for this to be made clear in Section 2. Further, these previous manuscripts provided data on hit %s and mass spectral resolution that could be used in the current manuscript to comment on how much each improvement made a difference to the overall performance of the HP-SPAMS.

**Answer:** Thank you for your suggestion. We have described the technical details in previous articles, and in Section 2 we have added references to descriptions of technical details in previous papers. And in section3, the results of the experiment are analyzed in terms of their relevance to technology.

3. Based on these previous manuscripts, the authors' main goal in the current manuscript appears to be the characterization of the full integrated system (HP-SPAMS) and comparison to the original SPAMS, rather than a detailed description of the "design", which is lacking, as noted above. The manuscript title needs to be revised to indicate this.

**Answer:** Thank you for your suggestion. Here we think "development" might be a little better than "design". Here we change the title to " Development and characterization of a High-Performance Single-Particle Aerosol Mass Spectrometer (HP-SPAMS)".

4. Several of the improvements made to this instrument have been previously described for other single-particle mass spectrometers, but this is not clear in the manuscript. This comment relates to the inadequate response of the authors to the previous requests for additional comparisons to non-SPAMS instruments. I provide examples (not meant to be comprehensive – the authors need to do a literature search) here. Delayed pulse extraction is used by several single-particle mass specs: miniSPLAT (Zelenyuk et al., 2015, J. Am. Soc. Mass Spec.), BAMS (Czerwieniec et al. 2005, J. Am. Soc. Mass Spec.), SPASS (Erdmann et al. 2005, Aerosol Sci. Technol.), LAMPAS-2 (Costa Vera et al. 2005, Rapid. Commun. Mass Spectrom.), ALABAMA (Clemen et al. 2020, AMT). Four-channel data acquisition is used on other single-particle mass spectrometers, including the ALABAMA (Brands et al. 2011, Aerosol Sci. Technol.) and ATOFMS (Pratt et al. 2009, Analytical Chem.). The 100 Hz Nd:YAG laser (Centurion, Quantel) is used on the Univ. of Michigan ATOFMS (Gunsch et al. 2017, Atmos. Chem. Phys.).

**Answer:** Thank you very much for your suggestion. We have cited a number of previous technical studies and analyzed them for relevance.

For example, we added information in introduction.

-line 80 "Delayed ion extraction could be an effective method to enhance the resolution of laser ionization mass spectrometry (Kinsel and Johnston, 1989), and this technique has been applied to instruments such as miniSPLAT (Zelenyuk et al., 2015), BAMS (Czerwieniec et al., 2005), ALABAMA (Clemen et al., 2020) and others. Our team has also developed an exponential form of ion delay extraction technique, which can effectively improve the performance of SPMS in terms of resolution and hit rate. (Chen et al., 2020; Li et al., 2018)."

- line 157."HP-SPAMS employed a 4-channel data acquisition technique, with positive or negative ion signals being acquired with two channels, the same technique that is used in the ALABAMA and ATOFMS instruments(Clemen et al., 2020; Pratt et al., 2009)."

5. The authors should cite Su et al. (2023, Atmos. Environ., "Analysis performance of single

particle aerosol mass spectrometer for accurate sizing and isotopic analysis of individual particles") as an additional study that characterized the SPAMS. In particular, that study also reported the accuracy of the 208Pb/206Pb ratio to be closer than observed for the SPAMS in the current work (Figure 10 in the current manuscript). This comparison should be addressed. In addition, it is not only the correlation that matters (discussed on Line 291 and shown in Figure 10), but also the comparison to the theoretical isotope ratio, which is not provided in the current manuscript.

**Answer:** Thank you very much for your suggestion. In fact, our manuscript submitted 02 Sep 2022, while Su et al. article received 12 January 2023, so it was not cited at that time. In addition, his experiments were precisely based on HP-SPAMS (Model 0535).

In order to better describe Figure 10, we calculated a deviation of 11.07% between experimental and theoretical Pb isotope ratios, which would be higher than in the study by Su et al. (2.6%). We believe that this could be due to (1) the fact that although we removed particles containing m/z 202 by sieving them out, there would still be interference from organic ions, whereas the study of Su et al. used a standard solution of elemental lead, which does not suffer from the problem of interference from organic ions, and (2) another possibility is that it could be influenced by the MCP detector, which is used to detect the lead isotopes in individual particles when they contain elemental lead at high enough levels due to the MCP's nonlinear gain saturation, which can lead to low measurements. For example, in the figure below, the isotopic ratio of lead is 2.157 when the signal is low, but when the signal strength is high, the isotopic ratio decreases to 1.835.

[Figure]

Here, we've revised the manuscript to state that, "In addition, the isotope ratio $^{208}Pb^+/^{206}Pb^+$ is 1.933, which is about 11.07% deviation from the theoretical isotope ratio for lead ($^{208}Pb^+/^{206}Pb^+$ = 2.174). In summary, the higher resolution of HP-SPAMS as well as the improved dynamic range of the data acquisition with peak intensities up to 20 V, which would make it more prominent for the use of isotopic identification, such as source apportionment or mineral identification (Marsden et al., 2018; Souto-Oliveira et al., 2018, Su et al.,2023). "

6. The authors state on Lines 38-39: "However, the limitations of current mass spectrometry (MS) detection capabilities render it not well suited for analyzing complex aerosol components in low

concentration environments." This is not accurate, as there have been single-particle MS measurements in the Arctic, free troposphere, and stratosphere – all low concentration environments. Only two references are cited in the next sentence, but there are many papers – especially using PALMS, SPLAT, and ATOFMS – that have shown highly successful data obtained in these environments.

**Answer:** Thank you very much for your suggestion, the expression here is indeed not quite accurate. What is meant to be expressed here is that it is not easy to obtain high temporal and spatial resolution at low concentrations, especially in environments such as on-board,shipboard or airborne. This has been changed to "However, due to the limitations of the SPMS detection capabilities, simultaneous detection with high temporal and spatial resolution in low concentration environments is not yet well realized."

7. On Lines 42-43 the authors refer to needing long-time collection and changing mixing state, but this is confusing because SPMS is, by definition, a real-time analysis method, such that long-time collection is not possible. Further, Pratt et al. (2009, Analytical. Chem.) shows aerosol mixing state with 2 and 4-min resolution. Therefore, it is clear that the authors' statement needs revision for multiple reasons.

**Answer:** Thank you for your suggestion. We have rewritten this part. "Although enough particles could can be obtained by long-time collection, this would result in a loss of temporal or spatial resolution, especially on airborne or shipboard instruments."

8. The authors state on Lines 44-45: "…the poor mass resolution and detection sensitivity of SPMS make it difficult to obtain an accurate analysis of aerosols. (Pratt et al. 2009; Zelenyuk et al. 2009)" Yet, neither of these papers shows this, and it is not an accurate statement, as there are many publications, especially using ATOFMS, SPLAT, and PALMS showing accurate, quantitative analysis by SPMS.

**Answer:** Thank you for your suggestion. We have rewritten this part. "Another reason is the low mass resolution and detection sensitivity of SPMS, leading to possible problems with the capability in terms of aerosol characterization, e.g., Liu et al. found in their study that vanadium markers ($V^+$ or $VO^+$) emitted from ships were interfered with by isobaric organic ions (Liu et al., 2017).

9. On Lines 56-57, the authors state "The particles that could be transmitted to the instrument are not entirely measurable, which could be related to the particle sizing efficiency and ionization probability", with no reference. This statement is confusing, as many published single-particle mass spec characterization studies have quantified transmission through multiple methods.

**Answer:** I'm sorry for the confusion. Here, we would like to express that even after the particles have passed through the inlet system, it is not always possible to be detected by the sizing system and ionized by the ionization laser. It has been changed into "The particles transmitted to the instrument through the inlet system are not fully measurable, which could also be related to the particle sizing efficiency and ionization probability(Gemayel et al., 2016)."

For quantification, as you say SPMS measurements do have some quantitative capability and

possibility, e.g. Rachel Gemayel et.al. (Development of an analytical methodology for obtaining quantitative mass concentrations from LAAP-ToF-MS measurements, Talanta,174,715-724,2019) investigated the correlation between LAAP-TOF and AMS measurements, however, this does not mean that the particles are fully ionized by the sizing system, e.g., the scattering efficiency is only 1-2%(Laser ablation aerosol particle time-of-flight mass spectrometry, Atmos. Meas. Tech. 9, 1947 -1959, 2016). Therefore, we believe that increasing the detection efficiency can improve the accuracy of SPMS analysis results.

10. On Lines 80-81, the authors state "Although there are many studies to improve one aspect of the performance of SPMS, the instrument's overall performance is insufficient." Yet, there have been many improvements to single-particle mass spec instruments over the past three decades, with great success, and many high-performing instruments exist. Please revise this statement to refer to the need to improve the SPAMS itself. Further, this statement makes it clear that a paragraph about the SPAMS itself is needed in Section 1, as the main goal of the current work is to compare the SPAMS and HP-SPAMS.

**Answer:** Thank you for your suggestion. It has been changed into " Although there have been several  studies based on previous SPAMS(Li et al., 2011) to improve the performance, e.g., resolution (Li et al., 2018), hit rate (Chen et al., 2020), mass accuracy (Chudinov et al., 2019; Zhu et al., 2020), etc., and the high-performance single-particle aerosol mass spectrometer (HP-SPAMS) has been applied to the study of particle density (Peng et al., 2021), diesel vehicle exhaust (Su et al., 2021a), and sea salts (Su et al., 2021b), the overall design of HP-SPAMS has not yet been described and characterized. In this study, the performance of HP-SPAMS will be described in detail, as well as a comparison of the performance with SPAMS. Firstly, the structure and design of the HP-SPAMS are described. Then the detection capability of HP-SPAMS and SPAMS for the number of particles is compared and analyzed in terms of the efficiency of sizing and hit rate. Furthermore, the detection results of the system as it is configured are shown. Finally, the improvement of resolution and sensitivity on the detection results of individual particles by HP-SPAMS and SPAMS are compared and examined."

11. Line 94: Provide the height with the inlet included as well.
**Answer:** Thank you for your suggestion. Here it has been modified to "960mm × 740mm × 1550mm".

12. Line 95: Provide the actual, rather than approximate weight.
**Answer:** Thank you for your suggestion. Here it has been changed "it weighs 220 kg"

13. On Lines 101-102, it is stated "…most particles have entered the separation cone." Please provide data, and quantify "most".

**Answer:** Thank you for your suggestion. Here, we specifically modify "most" to " Due to the difference in inertia between the gas and the particles, 50% of 50nm particles and 100% of 100nm-1μm particles have entered the separation cone. " in the manuscript and cite Zhuo's literature.In fact, the design and performance of APCs have been described in detail in Zhuo et al. (Improved Aerodynamic Particle Concentrator for Single Particle Aerosol Mass Spectrometry: A Simulation and Characterization Study, Chinese Journal of Vacuum Science and Technology, 41, 443-449, doi:10.13922/j.cnki.cjvst.202008026, 2021.)

14. Lines 106-107 state "…could theoretically transport particles in the range of 100-5000 nm." Where is the theoretical transmission curve data? Further, scattering efficiency data appears to only go out to ~2-3 um in Figure 4.

**Answer:** Thank you for your suggestion, here we added the theoretical transmission curve (Figure S1) in the supporting information.

(1) The theoretical transmission efficiency curves of ADLs are as follows, and the theoretical simulated values are in the range of 100nm-5000nm, and the transmission efficiency is better than 70%. And in the experiment, we can measure the particles of 150nm-5400nm by using the sizing system of the instrument, as shown in the figure.

(2) For Fig. 4 in the manuscript, the measurements show an upper limit of 3 μm due to the CPC (TSI 3775), so only particles up to 3 μm were measured for scattering efficiency.

[Figure]

[Figure]

15. Lines 112-113 state "Compared to the SPAMS, increasing the laser power enhanced the intensity of the scattered light from the particles." Please state how much the laser power was increased by, as well as provide data comparing the scattered light.

**Answer:** Thank you for your suggestion. It has been changed into"Compared to SPAMS, the laser power is increased from 75 mW to a maximum of 500 mW, which enhances the intensity of the light scattered by the particles."

The following figure shows the results of the response-averaged intensity of the PMT signal of PSL spheres at 230 nm versus the laser power. From the figure, increasing the power of the laser increases the signal-to-noise ratio of the scattered light signal within a certain power range.

[Figure]

16. Lines 114-115 state "the background noise level was effectively reduced, and the detection capability for small-size particles was improved." This is referred to again on Lines 190-192 with the same phrasing. How much was the noise level reduced? What size is "small"? How much was the detection of these "small-size" particles improved? Provide quantitative data.

**Answer:** Thanks. (1)The background noise level has changed from -9.8±6.4mV to -10±0.8mV, which indicates that the low-pass filtering reduces the influence of stray light. (2)The statement of "and the detection capability for small-size particles was improved." is not precise enough here. The point we are trying to make is that there is an improvement in the signal-to-noise ratio of the PMT, which correlates with both small and large particles that scatter light weakly. Here we have changed it into "the background noise level has changed from -9.8±6.4mV to -10±0.8mV, which indicates that the low-pass filtering reduces the influence of stray light"

[Figure]

17. Line 125 states that "laser beam homogenization" is employed, but I could not find a description of how this was done. The reference (Steele et al. 2005) is to a different single-particle mass spectrometer. Please clarify how this was set-up for the HP-SPAMS and that is an improvement over the SPAMS. Also, it is confusing when, on Lines 212-213, the authors refer to a "non-uniform Gaussian beam", suggesting that laser beam homogenization may not have been done? But, then Lines 255-256 mention a "uniform laser beam". So, the laser set-up is not clear.
**Answer:**
(1)I'm sorry to confuse you, the Centurion Plus laser is used in HP-SPAMS, and the spot is already shaped with a homogenized beam, as shown in the follow figure;

[Figure]

(2)Here, we have revised it into "Compared to the ionized laser of SPAMS, the laser of HP-SPAMS contains two features. One is that the laser frequency has been increased from 20 Hz in

SPAMS to 100 Hz. The second is that SPAMS employs a laser with a Gaussian beam, while HP-SPAMS employs a laser with a homogenized spot. The laser beam homogenization gives the spot approximate energy density at different locations, which was shown to improve the repeatability of measurements by Steel e's research"

(3)In lines 212-213, this is for the SPAMS Gaussian spot. Here it is modified to "Even for the same particle, differences in energy density at the edges and center of the SPAMS non-uniform Gaussian beam could cause the particles were not ionized." In lines 255-256, this refers to the fact that the laser beam of HP-SPAMS is a homogeneous.

18. Lines 161-162: It is stated "To compare the detection capability of the instrument at low concentrations, the fraction of the samples fed by the SPAMS and HP-SPAMS was diluted by 40 times…". Please provide the initial and diluted number concentrations, to provide important context for what is meant by "low".

**Answer:** Thank you for your reminder. It has been changed into " In order to compare the detection capabilities of the two instruments at low concentrations (1-5 μg/m3), laboratory air samples collected by SPAMS and HP-SPAMS were diluted 40-fold with an aerosol diluter from 60-180 μg/m3."

19. Please provide the current and previous critical orifice sizes for context. The reference here to Cahill et al. (2014) is for a different single-particle mass spectrometer.

**Answer:** Thank you for your reminder. The critical orifice sizes has been described in section 2.1 " To increase the inlet flow, HP-SPAMS increased the critical orifice diameter from 0.1 mm to 0.18-0.22 mm, increasing the inlet flux to 0.3-0.5 L/min, which is greater than the 0.1 L/min of SPAMS."

20. Lines 210-212: Here it is stated that particles of different composition require different laser energy thresholds, but hit % data is only provided for the ambient data (Figure 5) and is not provided for the four different particle types shown in Figure 4, for which chemically-dependent scattering efficiency data are shown. Add hit % data that corresponds to the lab scattering data in Figure 4, as presumably the entire instrument was running during the experiment, such that hit data should be available. Increase the discussion of chemical biases here, including the role of the low chosen laser power (0.5 mJ).

**Answer:** Thank you for your suggestion.

(1) In Fig. 5, we show the data for ambient as well as for standard PSLs.

(2) It is very unfortunate that none of the four particles used during the test, except for PSLs, could be ionized by the 266 nm laser at 0.5 mJ/pulse, which may be due to the high ionization thresholds of the pure NaCl, $(NH_4)_2SO_4$, DOS particles. This phenomenon is also described in the study by D. S. Thomson et al. (Thresholds for Laser-Induced Ion Formation from Aerosols in a Vacuum Using Ultraviolet and Vacuum-Ultraviolet Laser Wavelengths, Aerosol Science and Technology, 26(6), 1997: 544-559).

(3) 0.5 mJ may not apply to all particle ionization energies, but this will reduces the fragmentation of organic molecules. Here we have added as "In addition, both HP-SPAMS and SPAMS operate

at laser energies of about 0.5 mJ/pulse during the experiments, and lower fluence can reduces fragmentation of organic molecules."

21. Report the HP-SPAMS data in terms of number concentration, rather than #/min, in Figure 6. Then in the text, compare to the SMPS number concentration in the same size range, to the HP-SPAMS number concentration, to enable a quantitative comparison. This is important because the authors are claiming very high scattering and detection efficiencies in Figures 4-5, and so, a quantitative comparison is needed here. Currently only correlation analysis was completed, at the bottom of page 7. It would also be more meaningful to examine the correlation with the SMPS concentration in the same size range where the HP-SPAMS has high scattering and detection efficiency. The scatter and hit rates of the instrument are also important to discuss to provide context for the instrument performance.

**Answer: Thank you for your suggestion.**

(1) Figure 6 has been modified to "# /cm3".

[Figure]

(2) The discussion of efficiency by particle size segments is a good suggestion, and we have tried to analyze similar data. The figure below shows the data measured by HP-SPAMS and SMPS with a statistical interval of 136 seconds averaged. Although this data is already available, we believe that it is incorrect to compare the particles in the different size bands of the two instruments because the particle sizes measured by the two instruments are not the same size. The relationship is (Alla Zelenyuk et al. From Agglomerates of Spheres to Irregularly Shaped Particles: Determination of Dynamic Shape Factors from Measurements of Mobility and Vacuum Aerodynamic Diameters):

$$\frac{d_{va}}{d_b} = \frac{\rho_P}{\rho_0} \frac{1}{\chi_v \chi_{t,\theta}} \frac{C_c(d_{va}\chi_v\rho_0/\rho_P)}{C_c(d_b)}$$

For ambient particles, both $\rho_P$ and $\chi_v$ are unknown, making comparisons difficult.

[Figure]

22. Line 235-236: I could not find where these data are provided. Please clarify what is meant by "detect" (hit?), and provide data. Is this an average value or a maximum?

**Answer:** Thank you for your reminder. Here 'detect' refers to the number of particles (hit number) that have particle mass spectrum and size. It has been changed into"Figure S4 shows the curve of the number of particles sizing and hit as a function of time. The results in Figure S4 also show that even in a low PM$_{2.5}$ concentration of 1 µg/m$^3$ , the instrument still detects an average of about 5.04 hit particle per second".

Here we have a mistake, previously the ratio of the average number of particles per second to the average PM$_{2.5}$ mass concentration was used, which can lead to an inaccurate calculated value the large fluctuations in PM$_{2.5}$ concentration.

$$p = \frac{\frac{total\ particle\ number}{24*3600\ s}}{average\ PM2.5\ concentration} = \frac{1002141/24/3600\ \#/s}{19.4\ \mu g/m^3} = 5.97\ ^{\#/s}/_{1\ \mu g/m^3}$$

In fact, it may be more accurate for us to calculate this through per-minute data. As shown below, by the per-minute signals.

$$\bar{p} = \frac{\sum_{i=1}^{n} p_i}{n} = 5.04\ ^{\#/s}/_{1\ \mu g/m^3}$$

Where $n$ is the total time expressed in minutes, and $p_i$ is the number of particles per minute after normalized PM2.5 concentration

23. Lines 251-256: This discussion suggests that the ion transmission above m/z 100 improved, or the detection of these ions improved. Page 9 points to the data acquisition board change (signal amplification), which makes sense, but it isn't mentioned here as a likely explanation.

**Answer:** Thank you for your suggestion. (1)For greater clarity, this is modified to "In addition, Figure 8 shows that the number of HP-SPAMS peaks was significantly higher than that of SPAMS, especially for m/z above 100, which indicates that more ions are detected by HP-SPAMS compared to SPAMS, and more information about the particle components is obtained." (2)There is no amplification of the signal using 4 channels of data acquisition. The increase in dynamic range does improve the detection of ions with peak heights less than 100mV and has little effect on the detection of ions greater than 100mV. Figure 8 shows ions with peak heights greater than 100mv, so the 4-channel acquisition is not the reason for more peak detection.

24. Please clarify the sentence on Lines 278-279. Are the authors referring in the second part to the increased detection of ions > m/z 100? Also, how does this factor of 47.8 compare to the expected based on the data in Figures 4 and 5?

**Answer:** Thanks. (1)It is probably not too well described here, so we have added a description of Figure 8 to support the point made here. "In addition, Figure 8 shows that the number of HP-SPAMS peaks was significantly higher than that of SPAMS, especially for m/z above 100, which indicates that HP-SPAMS detects more ions and has better particle component detection sensitivity than SPAMS. "(2)The 47.8-fold improvement is mainly affected by three aspects: ① the inlet flow rate is improved by a factor of 3. ② the scattering efficiency is improved by a factor of 2-3. ③ the strike rate is improved by a factor of 5.5.

Additional comments:
- Line 16: By "multiple homogeneous masses", do the authors mean "isobaric ions"?
**Answer:** Thank you for your good suggestion. "isobaric ions" is a specialized term.

- Line 28: Incomplete sentence
**Answer:** Thanks. Delete"In the case of atmospheric lead-containing particles"

- Line 30: Change "outstanding" to "improved"
**Answer:** Thanks, it has been changed.

- Lines 33-34: Add reference to a review.
**Answer:** Thank you for your suggestion. we have added

- Line 108: Is one laser used with its beam split, or are two lasers used? Please clarify.
**Answer:** Thank you, two lasers used. we have rewritten this part.

- Lines 116-117: I assume that authors are referring to the calibration of velocity to dva. Please clarify the description here.
**Answer:** Thank you for your reminder. $d_a = a + bt + ct^2 + dt^3$ is correct here. We find that the time-of-flight-size fit is better than the velocity -size fit over a wide range of particle sizes (150-4000 nm).

[Figure]

- Line 170: There appears to be a typo or English phrasing problem here.
**Answer:** Thanks, we have revised it.

- Line 205: I assume the authors mean "delayed ion extraction" rather than direct current extraction here?
**Answer:** Thanks, we have revised it.

- Line 238: Do the authors mean "nuclei" instead of "nuclear"?
**Answer:** Thanks, we have revised it.

- Lines 247-249: The authors should cite their previous paper here, as it showed a detailed analysis of this improved resolution and would support the statement made.

**Answer:** Thanks, we have added the reference.

- Lines 260-261: This statement is confusing. Please clarify.
**Answer:** Thanks, here we delete"since most of the accurate relative atomic masses of most of the metal ions are less than rounded integers."

- Line 273: Please add "compared to the SPAMS" to provide context for the statement "more sensitive and accurate".
**Answer:** Thanks, we have added it

- Line 290: Include figure # here.
**Answer:** Thank you for your reminder. We have revised it.

- Lines 296-278: Clarify in this sentence that these improvements are compared to the SPAMS.
**Answer:** Thank you for your suggestion. We have changed it.

- Line 307: This states 0.97 corresponds to R-square, but Line 221 states it is a Pearson correlation (i.e. r, not r2). Which is correct?
**Answer:** Thank you for your suggestion. We've corrected it.

- Lines 311-312: It is stated "although the sensitivity of HP-SPAMS to analyze individual particles cannot be quantified". I don't understand this statement, as this is what the authors have characterized in this paper.
**Answer:** Thank you for your suggestion. Here, we would like to explain it. In the manuscript, we did not analyze the sensitivity of individual particle detection, e.g., how much content of a certain composition can be detected in a particle by HP-SPAMS.

- Line 314: Why is this not attributed to the ion detection, rather than the ion generation?
**Answer:** Thank you for your suggestion. We have changed it.

- Figure 4: Please change "Detection Efficiency" to "Scattering Efficiency" on the plot y axes to match the caption revision.
**Answer:** Thanks for the reminder, we've already revised it.

- Figure 6: Change "variation curve" to "temporal plot". Is the DUST METER the PM2.5 concentration? Provide the size range of the SMPS for context. Explain the HP-SPAMS plot legend in the figure caption.
**Answer:** Thank you for your suggestion. We have changed and added it.

- Figure 9: The c & d x axis labels show "m/z (Da)", but the figure caption suggests that the plot is showing m/z 206 relative peak area. Please fix.
**Answer:** Thank you for your suggestion. We have corrected it.

- Line 496: Fix to 208Pb+ vs 206Pb+ to agree with figure.

**Answer:** Thanks,we've already revised it.

**Referee #5**

I have received this manuscript in the revised version, after two reviewers commented on the manuscript. The comments of the reviewers have been addressed, but I did not check that in detail. I read the revised version without the tracked changes. Overall, the manuscript is informative and interesting. The improvements of the SPAMS are clearly shown and explained. However, I was surprised that after one round of revisions there are still so many (albeit relatively small) unclarities in the manuscript. I have listed these points below under "specific comments". Although I rate this as "minor revision", I think that my comments listed below have to be considered before the manuscript can be published.

General comments:
I would prefer "detection efficiency" over "scattering efficiency". The expression "detection efficiency" has been used in previous SPMS papers (e.g. Clemen et al., 2020, Murphy 2007, Zelenyuk and Imre 2005). I have seen that you changed it during revision but I recommend to go back to "detection efficiency".
**Answer:** Thank you very much for your comments. The use of scattering detection can better distinguish between a single laser beam detected or two laser beams detected. Similarly, scattering efficiency is also adopted in some papers (R. Gemayel et al.,Laser ablation aerosol particle time-of-flight mass spectrometry, Atmos. Meas. Tech. 9, 1947 -1959, 2016).

Throughout the manuscript you use very often the word "could". It is not clear to me if you mean this as past tense (referring to previous works) or as subjunctive (as in "it might be that…"). I suggest using "can" if you refer to processes and effects that are known to occur.
**Answer:** Thank you for your suggestion. We've changed "could" to "can" in several places.

Specific comments: (line number refer to manuscript-version3):
Line 17 "The as-improved HP-SPAMS": replace by "the improved HP-SPAMS"
**Answer:** Thanks for your suggestions. We already revised it.

Line 27: "In the case of atmospheric lead-containing particles". This sentence is not complete

**Answer:** Deleted sentence"In the case of atmospheric lead-containing particles."

Line 31: remove "could"
**Answer:** Thanks, we have removed it.

Line 32-33: remove "an aerosol monitoring instrument" and add "e.g., " before Murphy, 2007
**Answer:** Thanks for your suggestions. We already revised it.

Line 35: Move the full stop (colon) after the references, same in line 39, 41, 43, etc.
**Answer:** Thanks, we have changed it.

Line 47: change to "Cahill et al. (2014) developed…" and remove reference at end of the sentence
**Answer:** Thanks, we have changed it.

Line 54/55: I don't understand the meaning of "The particles that could be transmitted to the instrument are not entirely measurable…"
**Answer:** I am very sorry to confuse you here. It means here that the particles are not always detected by the sizing system or mass spectrometry after they pass through the feed system into the vacuum. It has been changed into"The particles transmitted to the instrument through the inlet system that could be transmitted to the instrument are not entirely fully measurable, which could also be related to the particle sizing efficiency and ionization probability"

Line 72-74: "To improve the accuracy of qualitative analysis, the identification of ions in the spectra is improved by organic fragment ions, metal oxide ions, and isotopic ions (Tan et al., 2002). However, these methods have reduced applicability for ions with mass deviations less than 1 Da.":
Please be more precise here. Isotopes are separated by exactly one Da. To separate different ions on one nominal m/z value, a much better separation than one Da is required (as you later show in Table 2).
**Answer:** It is expressed here in the sense of determining another peak by the peak of the isotope. For example, if we have a peak at m/z 54, which could be $^{54}Fe^+$ or $C_4H_6^+$, then in the past it would have been possible to determine what ion m/z 54 was by recognizing the abundance of $^{56}Fe^+$.

Line 74 "Ions delay extraction": I think "delayed extraction" or "delayed ion extraction" is the common expression
**Answer :** Thanks to your suggestion, we have changed "Ions delay extraction" to "delayed ion extraction."

Line 78: "Although there are many studies to improve one aspect of the performance of SPMS, the instrument's overall performance is insufficient.": To which instrument do you refer here? This can't be said in general with respect to all SPMS types. Do you refer specifically to the SPAMS?
**Answer:** Thank you for your suggestion. We have rewrite this part "Although there have been several studies based on previous SPAMS(Li et al., 2011) to improve the performance, e.g., resolution(Li et al., 2018), hit rate(Chen et al., 2020), mass accuracy(Chudinov et al., 2019; Zhu et al., 2020), etc., and the high-performance single-particle aerosol mass spectrometer (HP-SPAMS) has been applied to the study of particle density(Peng et al., 2021), diesel vehicle exhaust(Su et al., 2021a), and sea salts(Su et al., 2021b), the overall design of HP-SPAMS has not yet been described and characterized. In this study, the performance of HP-SPAMS will be described in detail, as well as a comparison of the performance with SPAMS. Firstly, the structure and design of the HP-SPAMS are described. Then the detection capability of HP-SPAMS and SPAMS for the number of particles is compared and analyzed in terms of the efficiency of sizing and hit rate. Furthermore, the detection results of the system as it is configured are shown. Finally, the

improvement of resolution and sensitivity on the detection results of individual particles by HP-SPAMS and SPAMS are compared and examined."

Line 83: "Furthermore, the detection results of the as-configured system are shown": replace by "Furthermore, the detection results of the system as it is configured are shown"
**Answer:** Thanks, we have changed it.

Line 102: "Fluent" software? Please give company name.
**Answer:** Thanks, change here to "optimized using Fluent 2021 R2 software (ANSYS, Inc.)".

Line 102-103: Was the ADL newly designed or was the existing SPAMS lens optimized?
**Answer:** Thanks. We have changed it into" and the new aerodynamic lens apertures were optimized to 5.0 mm, 4.8 mm, 4.4 mm, 4.1 mm, and 3.9 mm."

Line 108-110: As far as I understand, there are two mirrors and two photomultipliers, right? Or do you detect the scattered light from both laser beams on one PMT? But that's not how it looks in Figure 1.

**Answer:** Thanks for your input, it can be confusing here, we've rewritten this part." Two lasers (500 mW, LaserWave, LWGL532 nm-500 mW) were used to generate two continuous-wave 532nm laser beams orthogonally spaced 6 cm apart, which were each focused by a plano-convex lens (PLCX-25.4-51.5-C-532, CVI Laser Optics) into a 300 $\mu$m-sized spot at the first focus of the ellipsoidal mirror."

Line 115 "compensation model": should read "mode", right?
**Answer:** Thanks, we've changed it.

Line 142-143: "the SPMS was usually a single particle single acquisition,…" I think here is something wrong, I don't get the meaning of this sentence.

**Answer:** Thank you for your suggestion. We have changed it to "Although it is theoretically possible to achieve such a dynamic range using a high bit data acquisition card, ADC acquisition cards for SPMS typically perform only one acquisition for a single particle, which made the ADC acquisition card unable to suppress noise by accumulating acquisitions and thus cannot achieve the theoretical dynamic range in practical applications."

Lines 158 -160: "In order to characterize the performance of the instrument, the scattering efficiency, hit rate associated with the instrument is defined. The scattering efficiency of particle is defined as the ratio of the response pulse count of the photomultiplier tube to the total number of injected particles": change to "…the scattering efficiency and the hit rate associated…"
**Answer:** Thanks to your suggestion. We have corrected it.

As I said before, I recommend to keep "detection efficiency". It just needs to be defined properly. Later (Line 174 and Fig. 4) the reader learns that you defined two detection efficiencies, one for PMT1 and one for PMT2. That should be explained here already.

**Answer:** Thank you for your suggestion. We felt that the use of "scattering efficiency" would better avoid ambiguity.

Line 163: How is N_sizing obtained?

**Answer:** Thank you for your suggestion. The sizing system measures for a particle, firstly this particle must pass through both sizing lasers and there is no particle catching up during this time. Secondly, the particle is not within the busy time of the ionizing laser.

Line 174: Ok, here (and in Fig. 4) it is clearly explained that there are two PMT and two detection efficiencies. This should be mentioned above in the definition of the detection efficiency.

**Answer:** Thank you for your suggestion. It has been changed into "scattering efficiency". And it defined as "where legend PMT1 and PMT2 represent the scattering efficiencies of the two photomultiplier tubes of SPAMS, respectively. HP-PMT1 and HP-PMT2 are the scattering efficiencies of the two photomultiplier tubes of HP-SPAMS scattering efficiency. The results showed that the scattering efficiency of HP-SPAMS was higher than that of SPAMS for all the investigated aerosol particles."

Lines 182 -189: Can you separate the effects of the ADC and the new ADL? For example, sinf the ADC on the original SPAMS? It would be interesting to see the transmission curve of the ADC alone.

**Answer:** Thank you for your good suggestion. In fact, the design and performance of APCs have been described in detail in Zhuo et al. (Improved Aerodynamic Particle Concentrator for Single Particle Aerosol Mass Spectrometry: A Simulation and Characterization Study, Chinese Journal of Vacuum Science and Technology, 41, 443-449, doi:10.13922/j.cnki.cjvst.202008026, 2021.) . The result is shown in the figure. We have added the reference in section 2.1" Due to the difference in inertia between the gas and the particles, 50% of 50nm particles and 100% of 100nm-1μm most particles have entered the separation cone (Zhuo et al., 2021)"

[Figure]

Line 194-195: "This is because the aerosol produced by the atomizer will be charged".
What do you mean by this? I think it is the general widening of the particle beam with lower sizes, such that more particles miss the ablation laser spot. If it is only a charge effect, then HP-SPAMS should not show it due to the delayed extraction.
**Answer:** Thank you for your suggestion.

Line 196-197: "On the contrary, HP-SPAMS showed a better hit rate due to the use of DC-extraction (Chen et al., 2020; Clemen et al., 2020)."
Clemen et al. (2020) did not use the HP-SPAMS. Please correct (e.g. "a similar finding was reported by Clemen et al. (2020)").
Furthermore: You defined "DC" as direct current in SPAMS, now you use "DC-extraction" for HP-SPAMS. Shouldn't it read "pulse delay extraction" here?
**Answer:** Thank you for your suggestion. We have changed it into "On the contrary, HP-SPAMS showed a better hit rate due to the use of delayed ion extraction (Chen et al., 2020), a similar finding was reported by Clemen et al. (Clemen et al., 2020)".

Lines 201-212, Table 1 and Figure 6: PMT1-2 refers to particles detected at both PMTs?
**Answer:** Thank you for your suggestion. Here HP-PMT1-2 refers to the number of particles passing through the two laser beams without "particle catch-up". We have added it.

Lines 214-219 and Figure 7: Dividing the SPAMS and HP-SPAMS curves by the SMPS curves should roughly correspond to the detection efficiency and it rate displayed in Figs 3 and 4, respectively. Did you test that?
Besides, I would choose a logarithmic scale for the diameter axis.
**Answer:** Thank you for your good suggestion.
(1) We have attempted to present the data collected by HP-SPAMS in SMPS grain size intervals and time intervals, see below. However, we have found that this comparison is incorrect because the two instruments measure different particle sizes; SMPS measures mobility diameter ($d_b$) while HP-SPAMS measures aerodynamic diameter ($d_{va}$), and the conversion between the two is related to the shape factor and density of the particles (Alla Zelenyuk et al. From Agglomerates of Spheres to Irregularly Shaped Particles: Determination of Dynamic Shape Factors from Measurements of Mobility and Vacuum Aerodynamic Diameters):

$$\frac{d_{va}}{d_b} = \frac{\rho_P}{\rho_0} \frac{1}{\chi_v \chi_{t,\theta}} \frac{C_c(d_{va}\chi_v\rho_0/\rho_P)}{C_c(d_b)}$$

(2) We have changed the coordinates of Figure 7 to logarithmic scale.

[Figure]

Line 228: "stratospheric cloud condensation nuclear": there are not so many clouds in the stratosphere, except for polar stratospheric clouds. I suggest "upper tropospheric cloud nuclei"
**Answer:** Thanks for your suggestion. We have corrected it.

Secton 3.2 and Figure 8: Fig 8 in this form is not very illustrative. I suggest to add a line showing averaged data (e.g. 10 m/z values averaged).
**Answer:** Thank you for your suggestion. We think that the original expression of the scatter is a better representation of the rich number of spectral peaks. The mean and variance are used as follows.

[Figure]

Table 2 and Figure 9: This is indeed a great improvement and the first time (to my knowledge) that a single particle instrument shows mass spectra that resolve such ions.
**Answer:** Thank you very much for your affirmation.

Lines 263-265 (Section 3.3):

What is m/z 202 (mercury?), why does it need to be excluded? If 206, 207 and 208 are present, doesn't this confirm the presence of lead? If the particles additionally contain mercury, they are still "lead-containing", right?

**Answer**: Thank you for your suggestion. This is to remove the interference of polymeric organics or PAH (see figure).

[Figure]

Figures:

Figure 4: It is good that you kept "detection efficiency" in the graph. Please make it consistent in the caption.

**Answer**: Thanks, we have corrected it.

Figure 5, Caption: "SPMAS" -> "SPAMS"

**Answer**: Thanks, we have corrected it.

Figure 10:

Instead of the upper plots, I would prefer mass spectra that show the peaks at 206, 207, and 208 clearly separated.

Lower plots: check x-axis: should read m/z 206

**Answer**:  Thank you for your suggestion. (1) We believe that this demonstrates peaks other than lead ions, indicating that HP-SPAMS is more sensitive than SPAMS for detecting particles of the same type. (2) we have corrected it.

References:

Clemen, H. C., Schneider, J., Klimach, T., Helleis, F., Köllner, F., Hünig, A., Rubach, F., Mertes, S., Wex, H., Stratmann, F., Welti, A., Kohl, R., Frank, F., and Borrmann, S.: Optimizing the detection, ablation, and ion extraction efficiency of a single-particle laser ablation mass spectrometer for application in environments with low aerosol particle concentrations, Atmos. Meas. Tech., 13, 5923-5953, 10.5194/amt-13-5923-2020, 2020.
Murphy, D. M.: The design of single particle laser mass spectrometers, Mass Spectrom. Rev., 26, 150-165, 10.1002/mas.20113, 2007.
Zelenyuk, A., and Imre, D.: Single particle laser ablation time-of-flight mass spectrometer: An introduction to SPLAT, Aerosol Sci. Technol., 39, 554-568, 10.1080/027868291009242, 2005.

**Answer:** we have checked it.